# Scatterbrain: Unifying Sparse and Low-rank Attention Approximation

**Beidi Chen**[*][†]**, Tri Dao**[*][†]**, Eric Winsor** [†]**, Zhao Song** [§]**, Atri Rudra** [‡]**, Christopher Ré** [†]

[†] Department of Computer Science, Stanford University
[§] Adobe Research
[‡] Department of Computer Science and Engineering, University at Buffalo, SUNY
{beidic,trid,winsor,chrismre}@stanford.edu, zsong@adobe.com, atri@buffalo.edu

## Abstract

Recent advances in efficient Transformers have exploited either the sparsity or low-rank properties of attention matrices to reduce the computational and memory bottlenecks of modeling long sequences. However, it is still challenging to balance the trade-off between model quality and efficiency to perform a one-size-fits-all approximation for different tasks. To better understand this trade-off, we observe that sparse and low-rank approximations excel in different regimes, determined by the softmax temperature in attention, and sparse + low-rank can outperform each individually. Inspired by the classical robust-PCA algorithm for sparse and low-rank decomposition, we propose Scatterbrain, a novel way to unify sparse (via locality sensitive hashing) and low-rank (via kernel feature map) attention for accurate and efficient approximation. The estimation is unbiased with provably low error. We empirically show that Scatterbrain can achieve $2.1\times$ lower error than baselines when serving as a drop-in replacement in BigGAN image generation and pre-trained T2T-ViT. On a pre-trained T2T Vision transformer, even without fine-tuning, Scatterbrain can reduce $98\%$ of attention memory at the cost of only $1\%$ drop in accuracy. We demonstrate Scatterbrain for end-to-end training with up to $4$ points better perplexity and 5 points better average accuracy than sparse or low-rank efficient transformers on language modeling and long-range-arena tasks.

## 1 Introduction

Transformer models [64] have been adapted in a wide variety of applications, including natural language processing [26, 7, 51], image processing [10, 48], and speech recognition [43]. Training large Transformers requires extensive computational and memory resources, especially when modeling long sequences, mainly due to the quadratic complexity (w.r.t. sequence length) in attention layers. Recent advances in efficient transformers [36, 17, 35, 66, 22] leverage attention approximation to overcome the bottleneck by approximating the attention matrices. However, it is challenging to find a robust approximation method that balances the efficiency-accuracy trade-off on a wide variety of tasks [58, 59].

We categorize most of the existing approaches for efficient attention matrix computation into two major groups: exploiting either the sparsity, e.g., Reformer [36], SMYRF [22], or low-rank properties of the attention matrices, e.g., Linformer [66], Linear Transformer [35], and Performer [17]. However, these techniques usually have different strengths and focus on the performance of specific tasks, so their approximations still cause accuracy degradation on many other tasks. For instance, according to a recent benchmark paper [58] and our experiments, low-rank-based attention might be less effective on hierarchically structured data or language modeling tasks, while sparse-based variants do not perform well on classification tasks.

---

[*]Equal contribution. Order determined by coin flip.

35th Conference on Neural Information Processing Systems (NeurIPS 2021).

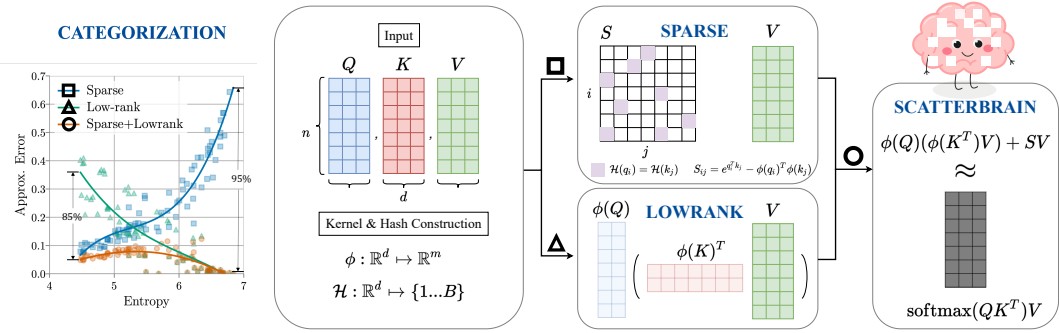

Figure 1: Left: regimes that sparse+low-rank approximation is more accurate, based on the entropy of the attention matrices. Right: Scatterbrain Workflow. For the attention layer in Transformers, after computing Query $Q$, Key $K$, and Value $V$ matrices, we approximate $\text{softmax}(QK^\top)V$ with two components: (i) sparse $SV$ (ii) low-rank $\phi(Q)(\phi(K)^\top V)$.

We observe that sparse and low-rank approximations are complementary for many attention matrices in practice, and sparse+low-rank could outperform each individually (Figure 1 left). We empirically categorize the regimes in which sparse or low-rank approximation achieves better error based on the softmax temperature of attention (of which the entropy of softmax distribution can be used as a proxy). We expect that sparse methods perform well if the attention depends on a few entries (low entropy softmax). In contrast, low-rank methods do better if the attention depends on a mixture of many components (high entropy softmax). This explains the phenomenon that current sparse and low-rank-based approaches excel on different kinds of tasks. A natural question is whether one could understand and unify the strength of both approaches. While it is NP-hard to find the optimal combination of sparse and low-rank approximations, Robust PCA [9] is a polynomial-time solution with tight approximation error. We observe that Robust PCA achieves lower approximation error than sparse or low-rank alone on attention matrices. The difference is most pronounced for "mid-range" entropy, where we observe that up to $95\%$ error reduction is possible.

The connection between Robust PCA and attention matrix estimation provides an opportunity to realize a more robust approximation. Specifically, given an attention matrix, one could adaptively perform sparse+low-rank approximation to obtain a low error. However, it comes with three challenges: (i) How to decompose the attention matrices into sparse and low-rank components and estimate them efficiently and accurately; Robust PCA is accurate but slow and requires materializing the full attention, while straightforward addition of sparse and low-rank attention will be inaccurate due to double counting. (ii) It is not clear if there is a theoretical guarantee that sparse + low-rank approximation is strictly better than sparse or low-rank in some regimes, though we observe the separation empirically. (iii) How does the lower approximation error transfer to end-to-end performance in real tasks.

In this paper, we propose Scatterbrain, an accurate and efficient robust estimation of attention matrices with theoretical guarantees to address the above challenges. Specifically:

- In Section 3, we observe that sparse and low-rank approximation are complementary and demonstrate that sparse + low-rank structure arises naturally when elements in the input sequence form clusters. We theoretically characterize and analyze the regimes where sparse, low-rank, and sparse+low-rank excel, dictated by the softmax temperature of attention.

- In Section 4, inspired by the classical Robust PCA algorithm, we propose Scatterbrain, which efficiently combines sparse and low-rank matrices to approximate attention. In particular, we use Locality Sensitive Hashing (LSH) to identify large entries of the attention matrix (after softmax) without materializing the full matrix and then leverage kernel approximation to parameterize the low-rank part. We prove that our method has a strictly lower approximation error than the low-rank baseline.

- In Section 5, we empirically validate our theory and the proposed method, showing that Scatterbrain accurately approximates the attention matrix, is memory efficient for long sequences, and works well across different tasks. First, we show that its approximation accuracy is close to our oracle Robust PCA and achieves $2.1\times$ lower error compared to other efficient baselines on real benchmarks. This leads to a direct application of Scatterbrain as a drop-in replacement to pre-trained full attention, thus reducing up to $98\%$ of the memory required for attention computations in pre-trained T2T-ViT and BigGAN while maintaining similar quality. Last we show that its superior accuracy and

efficiency can improve the efficiency-accuracy trade-offs of Transformer end-to-end training. On the WikiText-103 language modeling task, Scatterbrain achieves up to 1 point better perplexity compared to Reformer and Performer. On 5 benchmark long-range tasks, Scatterbrain improves the average accuracy by up to 5 points.[2]

## 2 Problem Setting and Related Work

We first define the approximation problem we aim to solve in this paper. Then we discuss the applications of sparse and low-rank techniques in efficient Transformers and introduce robust PCA algorithm.

**Problem Formulation:** In the attention matrix approximation problem, we are given three matrices, query, key, and value, $Q, K, V \in \mathbb{R}^{n \times d}$ to compute $\mathrm{softmax}(QK^\top)V$. We seek to reduce the quadratic complexity of $\mathrm{softmax}(QK^\top)$ (applied row-wise) with low approximation error. More precisely, for an approximation procedure $f$, we minimize two objectives, the approximation error $\mathbf{E}\big[\|f(Q,K) - \mathrm{softmax}(QK^\top)\|_F^2\big]$, and the computation/memory cost $\mathcal{C}(f(\cdot))$.

**Sparse, Low-rank Approximation for Attention Matrices:** Recent work exploits the sparsity patterns or finds a low-rank mapping of the original attention matrices to overcome the computational and memory bottlenecks in Transformers [36, 22, 54, 17, 35, 66]. Generally, we can divide most of the techniques into two categories – sparse and low-rank approximations. Reformer [36] is a representative sparse variant that uses LSH [3] to retrieve or detect the locations of the attention matrices with large values and reduce the computation from $O(n^2)$ to $O(n \log n)$. Performer [17] is an example of the low-rank variant, which uses kernelization to avoid explicit $O(n^2 d)$ computation. One problem of either the sparse or low-rank approximation is that the structure of the attention matrices varies in practice, and it is challenging to perform robust approximation on a wide range of attention matrices. For example, Wang et al. [66] observes that attentions tend to have more low-rank structures in lower layers and Ramsauer et al. [52] shows that they are sparser in the later stage of the training. Ideally, we want to unify the strength of both techniques, but it is NP-hard to find the best combination of sparse and low-rank approximation.

**Sparse + Low-rank and Robust PCA:** Fortunately, classical Robust PCA [9] presents a polynomial algorithm to find the approximately optimal or good combinations of sparse and low-rank approximation of the matrices. The sparse + low-rank matrix structure has been well studied in statistics and signal processing since the late 2000s [9]. This structure naturally generalizes low-rank [33, 63], and sparse [61] matrices. Scatterbrain is built on a line of work, e.g., Bigbird [71], Longformer [5] with the theme of combining multiple types of attention and another one in the optimal transport setting [37]. However, despite the multitude of papers, this sparse + low-rank matrix approximation has not been rigorously studied in the context of attention matrices. We undertake this study and show how we can relax the sparse + low-rank approximation from robust PCA, making it efficient while still retaining PCA's accuracy. In fact, our results shed further light on why Bigbird or Longformer work, as they are special cases of a single principled structure. An extended discussion of related work is in Appendix A.

## 3 Characterization of Sparse + Low-rank Approx. to Attention Matrices

We motivate the use of sparse + low-rank approximation of the attention matrices with the key observation that for many attention matrices, sparse and low-rank approximation are complementary, and their ideal combination (via Robust PCA) can outperform both (Section 3.1). Furthermore, we argue that the sparse + low-rank structure can arise naturally when elements in the input sequence form clusters, as dictated by the softmax temperature (Section 3.2).

### 3.1 Motivating Observations: Low-rank and Sparse Structures of Attention Matrices

We empirically characterize regimes where sparse and low-rank approximation are well-suited, based on the softmax temperature (for which we use the softmax distribution entropy is a proxy). Specifically, in Fig. 1 (left), we present the approximation error of the original attention matrices and the approximation (sparse or low-rank) of matrices sampled from a 4-layer Transformer trained on IMDb reviews classification [58]. We make two observations:

1. Sparse and low-rank approximation are complementary: sparse excels when the softmax temperature scale is low (i.e., low entropy), and low-rank excels when the softmax temperature is high (i.e., high entropy).

---

[2]Scatterbrain code is available at `https://github.com/HazyResearch/scatterbrain`

2. An ideal combination of sparse and low-rank (orange line in Fig. 1 left), obtained with robust PCA, can achieve lower error than both.

Similar observations on other benchmarks and details are presented in Appendix B.

### 3.2 A Generative Model of How Sparse + Low-rank Structure Can Arise

Sparse + low-rank parameterization is more expressive than either sparse or low-rank alone. Indeed, in the Appendix, we construct a family of attention matrices to show the separation between the approximation capability of sparse + low-rank vs. sparse or low-rank alone: for an $n \times n$ attention matrix, sparse or low-rank alone requires a $O(n^2)$ parameters to get $\epsilon$ approximation error in Frobenius norm, while sparse + low-rank only requires $O(n)$ parameters.

Moreover, we argue here that sparse + low-rank is a natural candidate to approximate generic attention matrices. We describe a generative model

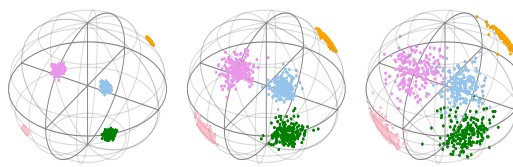

Figure 2: Visualization of the generative process, for three different values of the intra-cluster distance $\Delta$ (small, medium, and large). The vectors from the input sequence (rows of $Q$) form clusters that lie approximately on the unit sphere. Different colors represent different clusters.

of how the sparse + low-rank structure in attention matrices could arise when the elements of the input sequence form clusters. Under this process, we characterize how the softmax temperature dictates when we would need sparse, low-rank, or sparse + low-rank matrices to approximate the attention matrix. This result corroborates the observation in Section 3.1.

**Generative process of clustered elements in input sequence**  We describe here a generative model of an input sequence to attention, parameterized by the inverse temperature $\beta \in \mathbb{R}$ and the intra-cluster distance $\Delta \in \mathbb{R}$.

**Process 1.** *Let $Q \in \mathbb{R}^{n \times d}$, where $d \geq \Omega(\log^{3/2}(n))$, with every row of $Q$ generated randomly as follows:*

1. *For $C = \Omega(n)$, sample $C$ number of cluster centers $c_1,...,c_C \in \mathbb{R}^d$ independently from $\mathcal{N}(0, I_d/\sqrt{d})$.*
2. *For each cluster around $c_i$, sample $n_i = O(1)$ number of elements around $c_i$, of the form $z_{ij} = c_i + r_{ij}$ for $j = 1,...,n_i$ where $r_{ij} \sim \mathcal{N}(0, I_d \Delta/\sqrt{d})$. Assume that the total number of elements is $n = n_1 + \cdots + n_C$ and $\Delta \leq O(1/\log^{1/4} n)$.*

*Let $Q$ be the matrix whose rows are the vectors $z_{ij}$ where $i = 1,...,C$ and $j = 1,...,n_i$. Let $A = QQ^\top$ and let the attention matrix be $M_\beta = \exp(\beta \cdot A)$.*

We visualize this generative process in Fig. 2.

**Softmax temperature and approx. error**  We characterize when to use sparse, low-rank, or sparse + low-rank to approximate the attention matrices in Process 1, depending on the inverse temperature $\beta$. The intuition here is that the inverse temperature corresponds to the strength of interaction between the clusters. If $\beta$ is large, intra-cluster interaction dominates the attention matrix, the softmax distribution is peaked, and so we only need a sparse matrix to approximate the attention. If $\beta$ is small, then the inter-cluster attention is similar to intra-cluster attention, the softmax distribution is diffuse, and we can approximate it with a low-rank matrix. In the middle regime of $\beta$, we need the sparse part to cover the intra-cluster attention and the low-rank part to approximate the inter-cluster attention.

We formalize this intuition in Theorem 1 (in bounds below we think of $\epsilon$ as a constant). All the proofs are in Appendix D.

**Theorem 1.** *Let $M_\beta$, be the attention matrix in Process 1. Fix $\epsilon \in (0,1)$. Let $R \in \mathbb{R}^{n \times n}$ be a matrix. Consider low-rank, sparse, and sparse + low-rank approximations to $M_\beta$.*

1. **High temperature**: *Assume $\beta = o(\log n/\log d)$.*
    (a) **Low-rank**: *There exists $R$ with $n^{o(1)}$ rank (and hence $n^{1+o(1)}$ parameters) such that $\|M_\beta - R\|_F \leq \epsilon n$.*
    (b) **Sparse**: *If $R$ has sparsity $o(n^2)$, then $\|M_\beta - R\|_F \geq \Omega(n)$.*
2. **Mid temperature**: *Assume $(1 - \Delta^2)\log n \leq \beta \leq O(\log n)$.*

(a) ***Sparse + low-rank***: *There exists a sparse + low-rank $R$ with $n^{1+o(1)}$ parameters with $\|M_\beta - R\|_F \le \epsilon n$.*

(b) ***Low-rank***: *If $R$ is such that $n - \mathrm{rank}(R) = \Omega(n)$, then $\|M_\beta - R\|_F \ge \Omega(n)$.*

(c) ***Sparse***: *If $R$ has sparsity $o(n^2)$, then $\|M_\beta - R\|_F \ge \Omega(n)$.*

3. ***Low temperature***: *Assume $\beta = \Omega(\log n)$.*

   (a) ***Low-rank***: *If $R$ is such that $n - \mathrm{rank}(R) = \Omega(n)$, then $\|M_\beta - R\|_F \ge \Omega(e^{\beta(1-\Delta^2)})$.*

   (b) ***Sparse***: *There exists $R$ with sparsity $O(n)$ such that $\|M_\beta - R\|_F \le \epsilon \cdot e^{\beta(1-\Delta^2)}$*

## 4 Scatterbrain: Unifying Sparse and Low-rank Attention

We present Scatterbrain, and show that it approximates attention accurately and efficiently. Section 4.1 describes the challenges of designing an accurate and efficient approximation, and how obvious baselines such as Robust PCA or a simple combination of sparse attention and low-rank attention fail to meet both criteria. Section 4.2 demonstrates how Scatterbrain address the challenges (Fig. 1 contains a schematic of Scatterbrain). In Section 4.3, we show that Scatterbrain is unbiased with provably lower variance than low-rank baselines such as Performer.

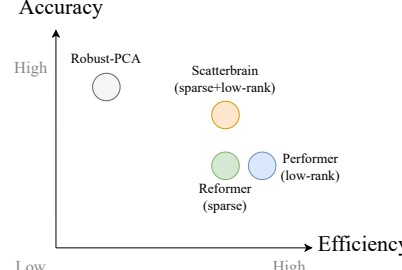

Figure 3: Qualitative comparison of approx. accuracy and efficiency, among Robust PCA, sparse (Reformer) and low-rank (Performer) attention, and Scatterbrain. Scatterbrain is more accurate while being efficient.

Fig. 3 shows a qualitative comparison between different methods of approximating the attention matrix: Robust PCA is accurate but slow, sparse (e.g., Reformer), and low-rank (e.g., Performer) attention are fast and memory-efficient but may not be very accurate, while Scatterbrain is more accurate than its sparse and low-rank counterparts while remaining just as efficient.

More details about the efficient implementation of Scatterbrain are in Appendix C.

### 4.1 Challenges of Designing an Accurate and Efficient Sparse + Low-rank Approximation

We seek a sparse + low-rank approximation of the attention matrix[3] $A$ that is *both* accurate and efficient. The natural theoretical baseline of Robust PCA is too slow and requires too much memory, while the most straightforward way of combining sparse attention and low-rank attention fails due to double counting on the support of the sparse attention.

1. If the goal is accuracy, Robust PCA is the most studied algorithm to find a sparse + low-rank approximation to a given matrix. It relaxes the NP-hard problem of finding the best sparse + low-rank approximation into a convex optimization problem, with the nuclear norm and $\ell_1$ constraints. Even though it can be solved in polynomial time, it is orders of magnitude too slow to be used in each iteration of a training loop. Moreover, it requires materializing the attention matrix, which defeats the main purpose of reducing compute and memory requirements.

2. On the other hand, one efficient way to get sparse + low-rank approximation of an attention matrix is to simply add the entries of a sparse approximation $S$ (say, from Reformer) and a low-rank approximation $\widetilde{Q}\widetilde{K}^\top$ for $\widetilde{Q}, \widetilde{K} \in \mathbb{R}^{n \times m}$ (say, from Performer). The sparse matrix $S$ typically has support determined randomly [16], by LSH [36, 22], or by clustering [54]. On the support of S, which likely includes the locations of the large entries of the attention matrix $A$, the entries of $S$ match those of $A$. One can multiply $(S + \widetilde{Q}\widetilde{K}^\top)V = SV + \widetilde{Q}(\widetilde{K}^\top V)$ efficiently because $S$ is sparse, and grouping $\widetilde{Q}(\widetilde{K}^\top V)$ reduces the matrix multiplication complexity when $m \ll n$, from $O(n^2 m)$ to $O(nmd)$. The approximation $S + \widetilde{Q}\widetilde{K}^\top$ matches $\widetilde{Q}\widetilde{K}^\top$ outside $\mathrm{supp}(S)$, hence it could be accurate there if $\widetilde{Q}\widetilde{K}^\top$ is accurate. However, $S + \widetilde{Q}\widetilde{K}^\top$ will not be accurate on the support of $S$ due to the contributions from both $S$ and from $\widetilde{Q}\widetilde{K}^\top$. Adjusting $\widetilde{Q}\widetilde{K}^\top$ to discount the contribution from $S$ is difficult, especially if we want to avoid materializing $\widetilde{Q}\widetilde{K}^\top$ for efficiency.

---

[3]For simplicity of discussion, we consider the unnormalized attention matrix $A = \exp(QK^\top)$, omitting the usual scaling of $\sqrt{d}$ and the softmax normalization constant.

## 4.2 Scatterbrain: Algorithm Intuition and Description

The simple insight behind our method is that on the support of the sparse matrix $S$, instead of trying to match the entries of the attention matrix $A$, we can set the entries of $S$ to discount the contribution from the low-rank part $\widetilde{Q}\widetilde{K}^\top$. This way, the approximation $S+\widetilde{Q}\widetilde{K}^\top$ will match $A$ exactly on the support of $S$, and will match $\widetilde{Q}\widetilde{K}^\top$ outside $\mathrm{supp}(S)$, which means it will still be accurate there if $\widetilde{Q}\widetilde{K}^\top$ is accurate. We do not need to materialize the full matrix $\widetilde{Q}\widetilde{K}^\top$ as need a subset of its entries is required, hence our approximation will be compute and memory efficient.

Scatterbrain thus proceeds in three steps: we construct a low-rank approximation $\widetilde{Q}\widetilde{K}^\top \approx A$, and construct a sparse matrix $S$ such that $S+\widetilde{Q}\widetilde{K}^\top$ matches $A$ on the support of $S$, then finally multiply $SV$ and $\widetilde{Q}(\widetilde{K}^\top V)$ and combine the result. More specifically:

1. **Low-rank Approximation**. We define a procedure LowRank that returns two matrices $\widetilde{Q},\widetilde{K} \in \mathbb{R}^{n\times m}$ such that $\widetilde{Q}\widetilde{K}^\top$ approximates $A$. In particular, we use a randomized kernel feature map $\phi\colon \mathbb{R}^d \to \mathbb{R}^m$ where $\phi(x) = \frac{1}{\sqrt{m}}\exp(Wx - \|x\|^2/2)$ with $W \in \mathbb{R}^{m\times d}$ randomly sampled, entry-wise, from the standard normal distribution $\mathcal{N}(0,1)$. We apply $\phi$ to each row vector of $Q,K$ matrices, and denote $\widetilde{Q}=\phi(Q)$ and $\widetilde{K}=\phi(K)$ (row-wise). Note that we do not materialize $\widetilde{Q}\widetilde{K}^\top$.
2. **Sparse Approximation**. We define a procedure SPARSE that returns a sparse matrix $S$ that matches $A-\widetilde{Q}\widetilde{K}^\top$ on $\mathrm{supp}(S)$. In particular, using a family of locality sensitive hash functions, compute the hash codes of each query and key vectors in $Q,K$ matrices (row-wise). Let $\mathcal{S}$ be the set of locations $(i,j)$ where $q_i$ and $k_j$ have the same hash codes (i.e, fall into the same hash bucket). Let $S$ be the sparse matrix whose support is $\mathcal{S}$, and for each $(i,j)\in\mathcal{S}$, define

$$S_{i,j} = \exp(q_i^\top k_j) - \phi(q_i)^\top \phi(k_j) = \exp(q_i^\top k_j) - \widetilde{q}_i^\top \widetilde{k}_j, \tag{1}$$

where $q_i, k_j, \widetilde{q}_i, \widetilde{k}_j$ are the $i$-th and $j$-th rows of $Q,K,\widetilde{Q},\widetilde{K}$ respectively. Note that we do not materialize $\widetilde{Q}\widetilde{K}^\top$.
3. **Scatterbrain Approximation**. With $\widetilde{Q},\widetilde{K}$ returned from LowRank and $S$ from SPARSE, we compute the (unnormalized) attention output with

$$\widetilde{O} = (\widetilde{Q}\widetilde{K}^\top + S)V = \widetilde{Q}(\widetilde{K}^\top V) + SV. \tag{2}$$

The precise algorithm, including the normalization step, as well as the causal/unidirectional variant, is described in Appendix C. We also note Scatterbrain's flexibility: it can use different kinds of low-rank and sparse approximation as its sub-components. The combination of Reformer and Performer is simply one instance of Scatterbrain. Instead of using Reformer as a sparse component, we could use local attention [5] or random block-sparse attention [16]. Instead of using Performer [17] as a low-rank component, we could also use Linear attention [35] or global tokens as in BigBird [71].

The Scatterbrain method would work exactly the same way. As long as the low-rank component is unbiased (e.g., Performer), its combination with any sparse component in Scatterbrain would yield an unbiased estimator of the attention matrix as shown below.

## 4.3 Scatterbrain: Analysis

Our method combines a low-rank approximation $\widetilde{Q}\widetilde{K}^\top$ (which has rank $m \ll n$) with a sparse approximation $S$. We argue that it is accurate (lower approximation error than baselines) and efficient (scaling the same as sparse or low-rank alone). The main insight of the analysis is that our approximation is exact for entries on the support of $S$ (picked by LSH), which are likely to be large. For entries not in the support of $S$ (likely to be small), our approximation matches the low-rank part (Performer) $\widetilde{Q}\widetilde{K}^\top$, which is unbiased and has low variance for these entries. As a result, Scatterbrain retains the unbiasedness of Performer [17] but with strictly lower variance.

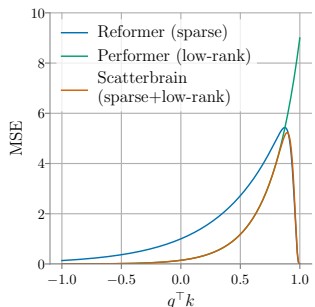

Figure 4: Per-entry MSE for different approximations, across a range of magnitude of $q^\top k$. Scatterbrain has low MSE for both small and large entries, thus outperforming its sparse (Reformer) and low-rank (Performer) counterparts.

We compare Scatterbrain to its low-rank baseline (Performer) and sparse baseline (Reformer). Performer is also based on the kernel approximation $\phi$, and simply uses $\widetilde{Q}\widetilde{K}^\top$ to approximate the attention matrix $A$. Reformer uses LSH to identify large entries of $A$, then compute a sparse matrix $S$ such that $S_{ij} = \exp(q_i^\top k_j)$ for $ij \in \text{supp}(S)$.

**Accuracy**: Because of the way $S$ is defined in Eq. (1), $\widetilde{Q}\widetilde{K}^\top + S$ matches $A = \exp(QK^\top)$ exactly on locations $(i,j) \in \mathcal{S}$, which are locations with likely large values. This addresses a weakness of low-rank methods (e.g., Performer) where the low-rank estimate is not accurate for locations with large values. We analyze the expectation and variance per entry of our estimator below (proof in Appendix D).

**Theorem 2.** *Define* $\sigma(q,k) = \exp(q^\top k)$, $\widehat{\sigma}^{\text{pfe}}$ *as Performer's estimator and* $\widehat{\sigma}^{\text{sbe}}$ *as Scatterbrain estimator. Denote* $\mathcal{S}^{d-1} \subset \mathbb{R}^d$ *as the unit sphere. Suppose* $q, k \in S^{d-1}$ *are such that* $\|q - k\| < \tau$. *Then:*

$$\mathbb{E}[\widehat{\sigma}^{\text{sbe}}(q,k)] = \sigma(q,k), \quad \text{Var}[\widehat{\sigma}^{\text{sbe}}(q,k)] = (1-p) \cdot \text{Var}[\widehat{\sigma}^{\text{pfe}}(q,k)] < \text{Var}[\widehat{\sigma}^{\text{pfe}}(q,k)] \quad (3)$$

*where* $p = \exp(-\frac{\tau^2}{4-\tau^2}\ln d - O_\tau(\ln\ln d))$.

Hence Scatterbrain is unbiased, similar to Performer [17], but with strictly lower variance. The variance is small if $\exp(q^\top k)$ is small (since $\text{Var}(\widehat{\sigma}^{\text{pfe}}(q,k))$ will be small), or if $\exp(q^\top k)$ is large (since the probability of not being selected by LSH, $1-p$, will be small). In Fig. 4, we plot the per-entry MSE of different methods from Theorem 2 when approximating the unnormalized softmax attention $\exp(QK^\top)$. Scatterbrain can approximate well both small entries (similar to the low-rank baseline, Performer), as well as large entries (similar to the sparse baseline, Reformer). Thus Scatterbrain has much lower MSE than Performer for large entries, and lower MSE than Reformer for small entries.

**Efficiency**: In Eq. (2), the computation $SV$ is efficient because $S$ is sparse, and $\widetilde{Q}(\widetilde{K}^\top V)$ is efficient because of the way we associate matrix multiplication (scaling as $O(nmd)$ instead of $O(n^2d)$, which is much bigger if $m \ll n$).

We validate these two properties of our approach in Section 5.

## 5 Experiments

We validate three claims that suggest Scatterbrain provides an accurate and efficient approximation to attention matrices, allowing it to outperform its sparse and low-rank baselines on benchmark datasets.

- In Section 5.1, we evaluate the approximation error and testing accuracy of different approximation methods on pre-trained models such as BigGAN and Vision Transformer. We show that the approximation by Scatterbrain is close to the Robust PCA oracle and up to $2.1\times$ lower approximation error than other efficient baselines.
- In Section 5.2, we validate that when trained end-to-end, Scatterbrain outperforms baselines (sparse or low-rank attention) on a wide variety of benchmark tasks, including language modeling, classification, and the Long-range Arena (LRA) benchmarks. Scatterbrain achieves up to 5 points higher average accuracy on the LRA benchmark compared to Performer and Reformer.
- In Section 5.3, we demonstrate the scalability of Scatterbrain, showing that it has comparable memory and time usage with simpler baselines (sparse or low-rank alone) across a range of input sequence lengths (Section 5.3), while requiring up to $12\times$ smaller memory than full attention.

All details (hyperparameters, data splits, etc.), along with additional experiments, are in Appendix E.

### 5.1 Scatterbrain's Approximation Accuracy

We evaluate Scatterbrain's approximation accuracy in three steps: (1) compare it with of Robust PCA (sparse+low-rank), our theoretical foundation and oracle (2) compare it with SMYRF[4] [22], Performer [17], which are popular variants of sparse and low-rank approximation to attention respectively and a naive baseline that directly adds SMYRF and Performer, (3) evaluate the inference accuracy when replacing full attention with Scatterbrain

Table 1: Top-1 Accuracy of pre-trained T2T Vision Transformer on ImageNet with different attention replacements. Error represents the average normalized approximation error to full attention.

| Attention | Top-1 Acc | Error (avg) |
|---|---|---|
| Full Attention | 81.7% | - |
| SMYRF | 79.8% | 11.4% |
| Performer | 80.1% | 7.5% |
| Baseline SMYRF + Performer | 79.7% | 12.6% |
| Scatterbrain | **80.7**% | **5.3**% |

---

[4]SMYRF is a variant of Reformer that does not require the key and query to be the same, which is necessary for experiments in this section.

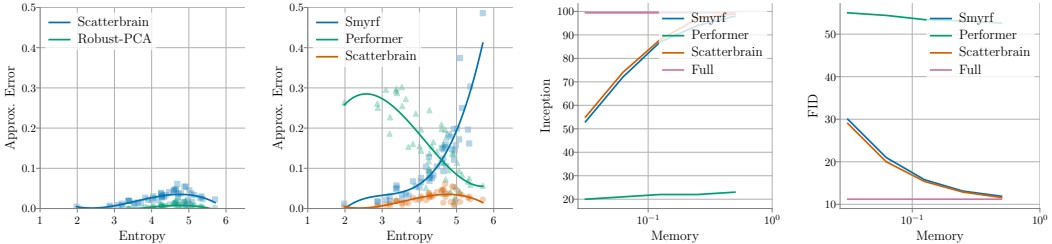

Figure 5: First: approximation comparison between Scatterbrain and its "lowerbound" Robust PCA. Second: comparison of error vs. entropy among SMYRF, Performer and Scatterbrain, three representatives of sparse, low-rank and sparse+low-rank approximations. Third and forth: Inception score (higher is better) and FID score (lower is better) of different attention variants for pretrained BigGAN.

approximation. Scatterbrain achieves error
within 20% of the oracle robust PCA, and up to $2.1\times$ lower error than SMYRF and Performer. When serving as a drop-in replacement for full attention, even without training, Scatterbrain can reduce the attention memory of Vision Transformer by 98% at the cost of only 0.8% drop of accuracy.

**Setup:** We use the attention matrices from pre-trained BigGAN and T2T-ViT. BigGAN is a state-of-the-art model in Image Generation for ImageNet. BigGAN has a single attention layer at resolution $64 \times 64$ (4096 queries). T2T-ViT has 14 attention layers. Scatterbrain sets the ratio between SMYRF and Performer based on the entropy of an observed subset of attention matrices in different layers. We allocate more memory to the low-rank component compared to the sparse part if the entropy is high.

**Scatterbrain and Robust PCA:** We first show that Scatterbrain approximates pre-trained attention matrices $10^5\times$ faster while its approximation error is within 20% on average. We also provide an example visualization on 100 attention matrices from the BigGAN generation process in Figure 5 (left).

**Scatterbrain vs. SMYRF and Performer:** We show that Scatterbrain approximates pre-trained dense attention matrices with very low error compared to sparse (Reformer) or low-rank (Performer). Measuring Frobenius approx. error on the BigGAN image generation task, Scatterbrain achieves $2\times$ lower error compared to Performer.

**Drop-in replacement for full attention:** We show that accurate approximation directly leads to efficient Inference. We replace BigGAN's dense attention with a Scatterbrain layer without other modifications. In 5 (right two), we show Inception and FID scores for Scatterbrain and other baselines under different memory budgets. Similarly, we use T2T-ViT [70], which is a token-to-token vision Transformer pre-trained on ImageNet [25]. In Table 1, we show the average approximation error of Scatterbrain for each layer and the end-to-end testing accuracy after replacing full attention with Scatterbrain and other baselines. Notably, Scatterbrain achieves 80.7% Top-1 accuracy, which is only 1% drop from the original 81.7% by full attention reducing up to 98% of the memory usage.

## 5.2 End-to-end Training Performance

Scatterbrain's accurate approximation of attention matrices allows it to outperform other efficient Transformer methods on benchmark tasks. Across a range of diverse tasks, both commonly used autoregressive tasks (sequence modeling) and benchmark long-range classification tasks (Long-Range Arena), Scatterbrain outperforms Performer (low-rank baseline) and Reformer (sparse baseline) by up to 4 points.

### 5.2.1 Auto-regressive Tasks

On the standard language modeling task of Wikitext-103, Scatterbrain obtains 1 point better perplexity than Reformer (sparse baseline), coming within 1.5 points of full attention.

**Settings:** We compare the performance of Scatterbrain against Reformer and Performer on one popular synthetic task, Copy, and one large language modeling task: WikiText103 [46]. Reformer is a representative sparse-approximation-based variant and Performer is a low-rank-approximation-based variant. The base model is vanilla Transformer [64]. We observed that generally allocating more memory budget to sparse tends to perform better, so Scatterbrain sets the ratio to 3:1 (sparse: low-rank component) for simplicity. The statistics of each dataset and model hyper-parameters are in Appendix E. We report the best results of each method in perplexity.

Table 2: The performance of Scatterbrain, Reformer, Performer and Full-Attention on Long-Range-Arena benchmarks and 2 popular language modeling tasks. We fix the same number of parameters (1/8 of the full) used for approximating the attention matrix for each method.

| Attention | Copy (ppl) | WikiText-103 (ppl) |
| --- | --- | --- |
| Full Attention | 1 | 25.258 |
| Reformer | 6.8 | 27.68 |
| Performer | 49 | 66 |
| Scatterbrain | **2.58** | **26.72** |

| Attention | ListOps | Text | Retrieval | Image | Pathfinder | Avg |
| --- | --- | --- | --- | --- | --- | --- |
| Full Attention | 38.2 | 63.29 | 80.85 | 41.78 | 73.98 | 59.62 |
| Reformer | 36.85 | 58.12 | 78.36 | 28.3 | 67.95 | 53.9 |
| Performer | 35.75 | 62.36 | 78.83 | 39.71 | 68.6 | 57.05 |
| Scatterbrain | **38.6** | **64.55** | **80.22** | **43.65** | 69.91 | **59.38** |

**Results:** Table 2 shows the testing perplexity for Scatterbrain and other baselines under the same parameter budget (each approximation is only allowed to compute $\frac{1}{8}$ of the full computation). Scatterbrain achieves comparable perplexity compared to the full attention Transformer model on Copy, and WikiText-103. Notably, Scatterbrain achieves 4 points lower perplexity on Copy and 1 point lower on WikiText-103 compared to Reformer, while Performer does not train stably on auto-regressive tasks (loss does not go down).

**Analysis:** We also analyze the results by visualizing the error of Reformer (sparse), Performer (low-rank), and Scatterbrain (sparse + low-rank) given the same number of parameters when approximating the full attention matrices for each attention layer during training (Appendix E). The conclusion is for language modeling tasks, sparse+low-rank has the smallest approximation error in most of the cases, and sparse has the largest error, which matches with the end-to-end results. It also confirms the observation in the popular benchmark paper [58] that kernel or low-rank based approximations are less effective for hierarchical structured data.

### 5.2.2 Classification Tasks

On a suite of long-range benchmark tasks (Long Range Area), Scatterbrain outperforms Reformer (sparse baseline) and Performer (low-rank baseline) by up to 5 points on average.

**Settings:** We compare the performance of Scatterbrain against Reformer and Performer on ListOps, two classifications: byte-level IMDb reviews text classification, image classification on sequences of pixels, a text retrieval, and pathfinder tasks. The datasets are obtained from the Long Range Arena (LRA) Benchmark [58], which is a recent popular benchmark designed for testing efficient Transformers. Similar to the auto-regressive tasks above, we use Reformer and Performer as baselines. The base model is also a vanilla Transformer. We follow the evaluation protocol from [58]. We report the best accuracy of each method.

**Results:** Table 2 shows the individual and average accuracy of each task for Scatterbrain and other baselines under the same parameters budget. Specially, each approximation is only allowed to use $12.5\%$ of the full computation. We can see Scatterbrain is very close to full attention even with a large reduction in computation and memory. Further more, it outperforms all the other baselines consistently on every task and achieves more than 5 point average accuracy improvement than sparse-based approximation Reformer and more than 2 point average accuracy improvement than low-rank-based variant Performer.

**Analysis:** Similarly, in order to analyze the performance of Reformer, Performer and Scatterbrain, we visualize their approximation error given the same number of parameters when approximating the full attention matrices for each attention layer during training (Appendix E). We again find that Scatterbrain has the smallest approximation error, while Performer is the worst on ListOps and Reformer has the largest error on classification tasks, which matches with

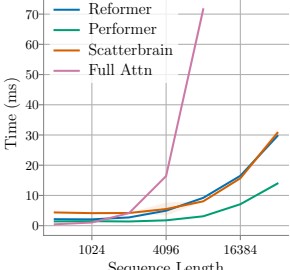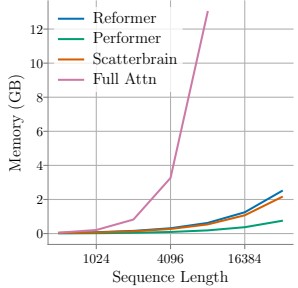

Figure 6: Speed and memory required by different efficient attention methods. Scatterbrain is competitive with SMYRF (sparse baseline) and Performer (low-rank baseline), while up to $3\times$ faster and $12\times$ more memory efficient than full attention for sequence length 4096.

the end-to-end results and confirms our observations earlier (sparse and low-rank approximation excel in different regimes).

### 5.3 Scatterbrain's Efficiency, Scaling with Input Sequence Length

We include ablation studies on the scalability of Scatterbrain in Fig. 6, showing that it is as computation and memory-efficient as simpler baselines such as SMYRF and Performer, while up to $3\times$ faster and $12\times$ more memory efficient than full attention for sequence length 4096. This demonstrates that our combination of sparse and low-rank inherits their efficiency.

We report run times and memory consumption of the sequence lengths ranging from 512 to 32768. We use a batch size of 16 for all runs and conduct experiments a V100 GPU. Since the efficiency would be largely conditioned on hardware and implementation details, we perform best-effort fair comparisons. We adapt the Pytorch implementation from `pytorch-fast-transformers` library for our baselines and implement Scatterbrain similarly without any customized cuda kernels.

## 6 Discussion

**Limitations.** As Scatterbrain has sparse attention as a component, it is not yet as hardware friendly (on GPUs and TPUs) as the low-rank component, which uses the very optimized dense matrix multiplication. This is the same limitation suffered by other sparse attention methods, but we are excited that more efficient sparse GPU kernels are being developed [31, 29].

**Potential negative societal impacts.** Our work seeks to understand the role of matrix approximation (and potentially energy savings) in the attention layer, which may improve a wide range of applications, each with their own potential benefits and harms. For example, making it language modeling more compute and memory efficient might facilitate spreading misinformation, and better image and video processing may make automatic surveillance easier. To mitigate these risks, one needs to address application-specific issues such as privacy and fairness, going beyond the error metrics we considered. Specially, for language models (LMs), while our work partially addresses the issue of environmental cost of LMs raised in [6], it does not address other issues such as unfathomable training data [6].

**Discussion and future work.** In this work, we make an observation on the sparse + low-rank structure of the attentions in Transformer models and theoretically characterize the regimes where sparse, low-rank and sparse + low-rank excel, based on the softmax temperature of the attention matrices. Motivated by this observation, we present Scatterbrain, a novel way to unify the strengths of both sparse and low-rank methods for accurate and efficient attention approximation with provable guarantees. We empirically verify the effectiveness of Scatterbrain on pretrained BigGAN, vision transformers, as well as end-to-end training of vanilla transformer. We anticipate that the study of this core approximation problem can prove useful in other contexts, such as generalized attention layers with other non-linearity beside softmax, and wide output layer in language modeling or extreme-classification.

### Acknowledgments

We thank Xun Huang, Sarah Hooper, Albert Gu, Ananya Kumar, Sen Wu, Trenton Chang, Megan Leszczynski, and Karan Goel for their helpful discussions and feedback on early drafts of the paper.

We gratefully acknowledge the support of NIH under No. U54EB020405 (Mobilize), NSF under Nos. CCF1763315 (Beyond Sparsity), CCF1563078 (Volume to Velocity), and 1937301 (RTML); ONR under No. N000141712266 (Unifying Weak Supervision); ONR N00014-20-1-2480: Understanding and Applying Non-Euclidean Geometry in Machine Learning; N000142012275 (NEPTUNE); the Moore Foundation, NXP, Xilinx, LETI-CEA, Intel, IBM, Microsoft, NEC, Toshiba, TSMC, ARM, Hitachi, BASF, Accenture, Ericsson, Qualcomm, Analog Devices, the Okawa Foundation, American Family Insurance, Google Cloud, Salesforce, Total, the HAI-AWS Cloud Credits for Research program, the Stanford Data Science Initiative (SDSI), and members of the Stanford DAWN project: Facebook, Google, and VMWare. The Mobilize Center is a Biomedical Technology Resource Center, funded by the NIH National Institute of Biomedical Imaging and Bioengineering through Grant P41EB027060. The U.S. Government is authorized to reproduce and distribute reprints for Governmental purposes notwithstanding any copyright notation thereon. Any opinions, findings, and conclusions or recommendations expressed in this material are those of the authors and do not necessarily reflect the views, policies, or endorsements, either expressed or implied, of NIH, ONR, or the U.S. Government. Atri Rudra's research is supported by NSF grant CCF-1763481.

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
