# Appendix

## Table of Contents

# A Extended Related Work

## A.1 Robust PCA

Robust Principle Component Analysis (robust PCA) is the problem of finding a composition of a matrix $M$ into a sum of sparse and low-rank components: $M = S + L$. It is a modification of PCA to accommodate corrupted observations (aka, noise). The sparse part covers the noise, while the low-rank part recovers the principle components. The most popular method to solve the problem is convex relaxation [7], where one minimizes the error $\|M - S - L\|_F^2$ subject to $\ell_1$ constraint on $\|S\|_1$ and nuclear norm constraint on $\|L\|_*$, in order to promote the sparsity of $S$ and the low-rankness of $L$. This convex problem can be solved with a variety of methods, such as interior point methods or the method of Augmented Lagrange Multipliers.

In our context, to find a sparse + low-rank decomposition of the attention matrix, one can also heuristically "peel off" the sparse part by finding the large entries of the attention matrix, then find a low-rank decomposition of the remainder. To avoid materializing the full attention matrix, one can use LSH to find potential locations of large entries, and use matrix completion [49] to find a low-rank decomposition. Gradient descent can find global optimum for this matrix completion problem [22]. However, it still requires too many iterations to be used in each training step.

## A.2 Efficient Transformers

**Sparse, Low-rank Approx.:** Transformer-based model such as BERT [35] has achieved unprecedented performance in natural language processing. Recently, Vision Transformers [27, 66] has also achieved comparable performance to the traditional convolutional neural network in computer vision tasks [63]. However, the quadratic computation of the attention layers constrains the scalability of Transformers. There are many existing directions to overcome this bottleneck, including attention matrix approximation such as Reformer [33], Performer [16], leveraging a side memory module that can access multiple tokens at once [53, 36, 35] such as Longformer [5] and BigBird [67], segment-based recurrence such as Transformer-XL [18] and Compressive Transformer [46]. Please refer to a recent survey [55] for more details. In this paper, we mainly explore within the scope of approximating dense or full attention matrices.

**Existing combination of Sparse and Low-rank Attention:** Our focus on the classical and well-defined problem of matrix approximation, as opposed to simply designing an efficient model that performs well on downstream tasks (e.g., Longformer, Luna, Long-short transformer, etc.) affords us several advantages: (i) Easier understanding and theoretical analysis (Section 3, 4). We see that Scatterbrain yields an unbiased estimate of the attention matrix, and we can also understand how its variance changes. (ii) Clear-cut evaluation based on approximation error, as well as the ability to directly replace a full attention layer with Scatterbrain attention without re-training (Section 5). This setting is increasingly important as transformer models are getting larger and training them from scratch has become prohibitively costly. Other methods such as Luna and Long-short transformer are not backward compatible with pre-trained models.

Here we compare Scatterbrain with other work mentioned by the reviewer, showing how most of them are special cases of Scatterbrain. We will also add this discussion in the updated version of the manuscript.

- Longformer [5]: a special case of Scatterbrain where the sparse component is local attention, and the low-rank component is the global tokens. Global tokens can be considered a restricted form of low-rank approximation.
- BigBird [67]: a special case of Scatterbrain where the sparse component is local + random sparse attention, and the low-rank component is the global tokens. The use of global tokens makes the model unsuited for autoregressive modeling. On the other hand, Scatterbrain's generality allows it to use other kinds of low-rank attention (e.g., Performer), and thus Scatterbrain works on both the causal/autoregressive and the bidirectional/non-causal attention settings. BigBird's motivation is also quite different from ours: they aim to design efficient attention such that the whole Transformer model is still a universal approximator and is Turing complete. Our goal is more concrete and easier to evaluate: we approximate the attention matrices, to get a small Frobenius error between the Scatterbrain attention and the full attention matrices.

- Luna [40] (concurrent work): they use a fixed-length extra sequence and two consecutive attention steps: the context sequence attends to the extra sequence, and then the query sequence attends to the extra sequence. This is similar in spirit to low-rank attention (Linformer) and global tokens, but it is not a low-rank approximation due to the non-linearity between the two attention steps. It is not clear to us that it combines different kinds of attention.
- Long-short transformer[68] (concurrent work): a special case of Scatterbrain where the sparse component is local attention and the low-rank component is Linformer.

## A.3 Locality Sensitive Hashing for Efficient Neural Network Training

Locality Sensitive Hashing (LSH) has been well-studied in approximate nearest-neighbor search [28, 31, 51, 2, 26, 10]. Since the brute-force approach for similarity search is computationally expensive, researchers have come up with various indexing structures to expedite the search process. Usually this comes with trade-offs on the search quality. Based on these indexing structures, one can achieve sub-linear search time. LSH has been used in estimation problem as well [12, 11].

Recently, there has been several work taking advantage of LSH data structures for efficient neural network training. During training process, the weight matrices are slowly modified via gradients derived from objective functions. If we consider the weights as the search data and input as queries, we can view neural network training as a similarity search problem. For example, [13, 17, 38] proposes an algorithm which performs sparse forward and backward computations via maximum inner product search during training. It is based on the observation that the model is usually over-parameterized so the activation for a given input could be sparse and LSH is used to find or impose the sparse structure. Similarly, LSH based algorithms have also been used in Transformers [13, 14], where LSH is used to capture the sparse structure of the attention matrices. They can largely reduce the memory bottleneck of self-attention modules especially over long sequences in Transformer. Though [14] has done some exploration to improve LSH accuracy-efficiency trade-offs through learnable LSH, most of the above works have limited understanding on when and where LSH can perform well.

## A.4 Structured Matrices for Efficient Machine Learning Models

Sparse + low-rank is an example of a class of structured matrices: those with asymptotically fast matrix-vector multiplication algorithm ($o(n^2)$ time complexity) and few parameters ($o(n^2)$ space complexity). Common examples include sparse, low-rank matrices, and matrices based on fast transforms (e.g., Fourier transform, circulant, Toeplitz, Legendre transform, Chebyshev transform, and more generally orthogonal polynomial transforms). These classes of matrices, and their generalization, have been used in machine learning to replace dense matrices in fully connected, convolutional, and recurrent layers [52, 58, 29]. De Sa et al. [23] shows that any structured matrix can be written as product of sparse matrices, and products of sparse matrices even with fixed sparsity pattern have been shown to be effective at parameterizing compressed models [19, 1, 20].

In our setting, it remains difficult to approximate the attention matrix with these more general classes of structured matrices. This is because many of them are fixed (e.g., Fourier transform, orthogonal polynomial transforms), and there lacks efficient algorithms to find the closest structured matrix to a given attention matrix.

# B Motivating Observations: Low-rank and Sparse Structures of Attention Matrices

We aim to build a deeper understanding of sparse and low-rank structures in real attention matrices: where each of them excel, and the potential for their combination. Specifically, we

- show that sparse and low-rank approximation errors are negatively correlated (through statistical tests),
- characterize regimes where each of sparse and low-rank approximation are well-suited, as dictated by the entropy of the softmax attention distribution, and
- demonstrate that sparse + low-rank has the potential to achieve better approximation than either.

## B.1 Setup

Denote $M$ as the attention matrix (after softmax) and $\mathcal{H}$ as entropy. We measure approximation error by the Frobenius norm or the original matrix and the approximation (sparse or low-rank). All the observed attention matrices in this section are from (1) a 4-layer vanilla Transformer trained from scratch on char-level IMDb reviews classification [54] (2) a 16-layer vanilla Transformer trained from scratch on WikiText103 [42] (3) a 1-layer (attention) pre-trained BigGAN on ImageNet [24]. To collect attention matrices for IMDb and WikiText103, we first save checkpoint of the models in every epoch; then evaluate 100 samples from validate data for each checkpoint and collect attention matrices from each layer each head. Note we take the median of the stats (error) for those 100 samples if it is difficult to visualize. To collect attention matrices for BigGAN, we generate 100 samples and collect the attention on the fly.

## B.2 Observation 1: Sparse and low-rank approximation errors are negatively correlated

Table 3: The Spearman's rank, Pearson and Kendall's Tau correlation coefficients between Sparse and Low-rank approx. error on IMDb, WikiText-103, and BigGAN-ImageNet. P-values of $< 0.05$ indicate statistical significance. The two errors are negatively correlated.

|  | IMDb | | WikiText103 | | BigGAN-ImageNet | |
|---|---|---|---|---|---|---|
|  | Coef | p-value | Coef | p-value | Coef | p-value |
| Spearman's rank | -0.89 | $< .00001$ | -0.63 | $< .00001$ | -0.21 | $< .00001$ |
| Pearson | -0.78 | $< .00001$ | -0.61 | $< .00001$ | -0.31 | $< .00001$ |
| Kendall's Tau | -0.74 | $< .00001$ | -0.51 | $< .00001$ | -0.18 | $< .00001$ |

We fixed the number of parameters, $K$, allowed for each attention matrix approximation and collect the errors from ideal sparse and low-rank approximations: top$-K$ entries for each row of the matrix for sparse and top$-K$ eigenvalues for low-rank. Then we run three standard statistical correlation tests [4, 56], Spearman, Pearson and Kendall's Tau on sparse and low-rank approximation error for all the matrices. We can see from Table 3 that errors are significantly negatively correlated (p-value $< 0.05$). Further more, the left three plots on Figure 7 visualizes the correlation between the two errors on three datasets.

This negative correlation suggests that there is some property of the softmax attention distribution which determines when sparse or low-rank excels. We validate this claim in the next observation.

## B.3 Observation 2: Sparse approximation error is lower when softmax entropy is low and low-rank approximation error is lower error when entropy is high

We visualize the sparse and low-rank approximation error against the entropy of attention matrices $\mathcal{H}(M)$ (applied to each row, then averaged) on the right plot in Figure 7. The attention matrices are $\in \mathbb{R}^{1024 \times 1024}$ (padded) so the x-axis has range from $[0, \ln(1024)]$. For high-entropy distributions (more diffused) low-rank matrices approximates the attention matrix well. For low-entropy distributions (more peaked), sparse matrices are better-suited.

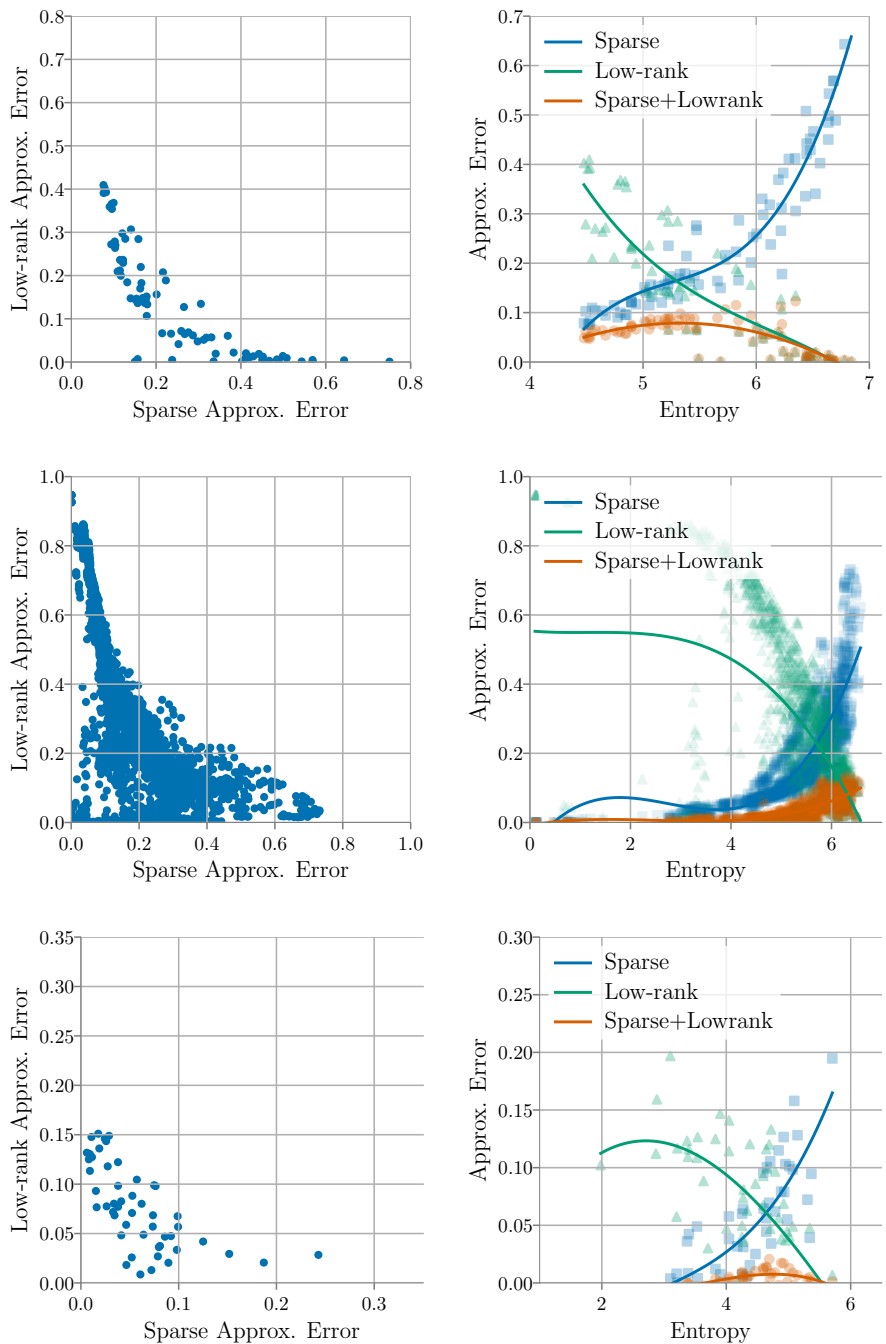

Figure 7: Characterization of the relationship between the softmax distribution of each attention matrix row and approximation error of sparse, low-rank and sparse+low-rank. The top, middle and bottom plots are for IMDb, WikiText103 and BigGAN-ImageNet respectively. Left: The approximation error of sparse and low-rank are negatively correlated. Sparse performs well when low-rank does not, and vice versa. Right: Entropy of the softmax attention distribution (i.e., scale of logits) determines the regimes where sparse, low-rank, or sparse + low-rank perform well. Sparse + low-rank yields better approximation than sparse or low-rank alone, across the board.

This implies that sparse and low-rank approximations could be complementary: if we can combine the strength of both, it is possible to come up with a better approximation across more general scenarios. Therefore, in the next observation, we try to combine sparse and low-rank approximations.

### B.4 Observation 3: Sparse + Low-rank achieves better approximation error than sparse or low-rank alone

We find an approximation of the attention matrix of the form $S + L$, where $S$ is sparse and $L$ is low-rank. This problem has a rich history and is commonly solved with Robust PCA. As shown in [7], across the range of entropy, sparse + low-rank approximation can achieve lower error than either sparse or low-rank when choosing the correct mix ratio of sparse and low rank approximation ideally (with robust-PCA).

Motivated by the fact that sparse and low-rank approximations of attention matrices have complementary strengths (Observations 1 and 2), one might want to combine them (Observation 3) in hope of yielding a more robust approximation that works well across different kinds of attention matrices. The above introduces three main challenges that we have addressed in the main paper:

- how to find sparse + low-rank decomposition of an attention matrix that is compute efficient (the most studied algorithm, robust PCA, is orders of magnitude too slow to be done at each training iteration) and memory efficient (i.e., without materializing the full matrix) (Section 4),
- if we can find such a sparse + low-rank decomposition, how accurate is the approximation (Section 4.3),
- how expressive is the sparse + low-rank parameterization, i.e., are there natural classes of matrices where sparse + low-rank yields asymptotically better approximation than sparse or low-rank alone) (Section 3)?

# C  Scatterbrain Algorithm and Implementation Details

Let $Q, K \in \mathbb{R}^{n \times d}$ be the query and key matrices respectively, and $V \in \mathbb{R}^{n \times d}$ be the value matrix. Let the rows of $Q$ be $q_1, \ldots, q_n$, and the rows of $K$ be $k_1, \ldots, k_n$. The attention computes:

$$\mathrm{softmax}(QK^\top)V,$$

with softmax applied row-wise, where for each vector $v \in \mathbb{R}^n$, $\mathrm{softmax}(v) = \frac{1}{\sum_{j=1}^n e^{v_j}} [e^{v_1}, \ldots, e^{v_n}]^\top$. Here we omit the usual scaling of $\frac{QK^\top}{\sqrt{d}}$ for simplicity since that could be folded into $Q$ or $K$. Note that $\mathrm{softmax}(QK^\top) = D^{-1} \exp(QK^\top)$, where the exponential function is applied element-wise and $D$ is a diagonal matrix containing the softmax normalization constants ($D_{i,i} = \sum_{j=1}^n \exp(q_i^\top k_j)$). Then attention has the form $D^{-1} \exp(QK^\top)V$.

We describe the Scatterbrain approximation algorithm in Algorithm 1. This includes the normalization step.

---

**Algorithm 1** Scatterbrain Approximation of Attention

---

1: **Input:** $Q, K, V \in \mathbb{R}^{n \times d}$, hyper-parameters $m, k, l$
2: **procedure** INIT($m, k, l$)
3:    Sample $W \in \mathbb{R}^{m \times d}$ where $W_i \sim \mathcal{N}(0, 1)$ i.i.d.
4:    Kernels $\phi \colon \mathbb{R}^d \mapsto \mathbb{R}^m$, $\phi(x) = \frac{\exp(Wx - \|x\|^2/2)}{\sqrt{m}}$
5:    Hash $\forall l \in [L]$, $\mathcal{H}_l = \{h_{l,k}\}_{k \in [K]}$, $\mathcal{H} = \cup_{l \in [L]} \mathcal{H}_l$
6: **end procedure**
7: **procedure** LOWRANKAPPROX($Q, K, V, \phi$)
8:    $\widetilde{Q} = \phi(Q)$, $\widetilde{K} = \phi(K)$                                                    ▷ applied to each row
9:    **return** $\widetilde{Q}(\widetilde{K}^\top V)$, $\widetilde{Q}(\widetilde{K}^\top)1_n$.
10: **end procedure**
11: **procedure** SPARSEAPPROX($Q, K, V, \phi, \mathcal{H}$)
12:    $\mathcal{S} = \{(i,j) | \mathcal{H}(Q_i) = \mathcal{H}(K_j)\}$
13:    $S \leftarrow$ sparse matrix whose support is $\mathcal{S}$
14:    **for** $(i,j) \in \mathcal{S}$ **do**
15:       $S_{ij} = \exp(q_i^\top k_j) - \phi(q_i)^\top \phi(k_j)$.
16:    **end for**
17:    **return** $SV$, $S1_n$.
18: **end procedure**
19: **procedure** SCATTERBRAINAPPROX($Q, K, V$)
20:    $\phi, h \leftarrow$ INIT($m, k, l$).
21:    $O_{\mathrm{lr}}, D_{\mathrm{lr}} \leftarrow$ LOWRANKAPPROX($Q, K, V, \phi$).
22:    $O_{\mathrm{s}}, D_{\mathrm{s}} \leftarrow$ SPARSEAPPROX($Q, K, V, \phi, h$).
23:    **return** $\mathrm{diag}(D_{\mathrm{lr}} + D_{\mathrm{s}})^{-1}(O_{\mathrm{lr}} + O_{\mathrm{s}})$.
24: **end procedure**

---

**Autoregressive / Causal / Unidirectional Attention**  To approximate autoregressive attention, we simply use the autoregressive variant of low-rank attention, and apply the autoregressive mask to the sparse attention. In particular, let $M \in \mathbb{R}^{n \times n}$ be the autoregressive mask, whose lower triangle is all ones and the rest of the entries are zero. The unnormalized attention matrix is $\exp((QK^\top) \odot M)$, and the unnormalized output is $\exp((QK^\top) \odot M)V$, where $\odot$ is elementwise multiplication.

The low-rank autoregressive variant computes $((\widetilde{Q}\widetilde{K}^\top) \odot M)V$, though with a custom GPU kernel / implementation so as not to materialize the $n \times n$ matrix. For the sparse component, we simply mask out locations $S_{ij}$ where $i > j$. That is, we can perform $S \odot M$ efficiently. As a result, we can compute the Scatterbrain output $((\widetilde{Q}\widetilde{K}^\top) \odot M)V + (S \odot M)V$ efficiently.

# D Proofs

## D.1 Expressiveness of Sparse + Low-rank Matrices

To motivate the use of sparse + low-rank matrices, we describe a family of attention matrices where sparse + low-rank matrices need asymptotically fewer parameters to approximate the attention matrix, compared to sparse or low-rank matrices alone. For there cases, either sparse or low-rank alone requires a quadratic number of parameters ($O(n^2)$, where $n \times n$ is the dimension of the attention matrix) to get $\epsilon$ approximation error in Frobenius norm, while sparse + low-rank only requires $O(n)$ parameters.

We construct a matrix family that shows the separation between the approximation capability of sparse + low-rank vs. sparse or low-rank alone. More specifically, we will use diagonal + low-rank (a special case of sparse + low-rank).

**Example 1.** *Let $\epsilon$ denote a parameter that satisfies $\epsilon \in (0, 1/2]$. Consider the following randomized construction of a matrix $Q \in \mathbb{R}^{n \times d}$ with $d \geq 6\epsilon^{-2} \log n$ and $d = \Theta(\epsilon^{-2} \log n)$, where each entry of $Q$ is picked independently and uniformly at random from $\{\pm 1/\sqrt{d}\}$. Let $M = \sigma(QQ^\top)$ where $\sigma$ is the elementwise exponential function (we first ignore the normalization term of softmax here).*

It can be shown (e.g. by Hoeffding's inequality) that with high probability

$$(QQ^\top)_{i,j} = \begin{cases} 1, & \text{if } i = j; \\ \in [-\epsilon, \epsilon], & \text{otherwise.} \end{cases}$$

Since $M = \sigma(QQ^\top)$ where $\sigma$ is the elementwise exponential function,

$$M_{i,j} = \begin{cases} e, & \text{if } i = j; \\ \in [1 - O(\epsilon), 1 + O(\epsilon)], & \text{otherwise.} \end{cases}$$

Intuitively, as the attention matrix $M$ has large diagonal entries, low-rank matrices will not be able to approximate it well. However, the off-diagonals are also of reasonable size, thus making sparse approximation difficult. With sparse + low-rank, we can use the sparse part to represent the diagonal, and the low-rank part to represent the remaining elements, allowing it to approximate this matrix well. We formalize this separation in the theorem below.

**Theorem 3.** *Let $M$ be the attention matrix from Example 1. For any $\gamma \in [0, 1]$, with probability at least $1 - n^{-1}$, there exists a sparse + low-rank estimator with $O(\gamma^{-1} n^{3/2} \log n)$ parameters that achieve $\gamma \sqrt{n}$ Frobenius error. For any matrix $R \in \mathbb{R}^{n \times n}$ with rank such that $n - \text{rank} = \Omega(n)$ (e.g., $R$ has $o(n^2)$ parameters), with probability at least $1 - n^{-1}$, we have $\|M - R\|_F \geq \Omega(\sqrt{n})$. Moreover, any matrix $E_S$ that has row sparsity $k$ (each row has less than $k$ non-zeros) such that $n - k = \omega(1)$ (e.g., $E_S$ has $o(n^2)$ parameters) will have error $\|M - E_S\|_F \geq \Omega(\sqrt{n})$ with probability at least $1 - n^{-1}$.*

We see that for any $\gamma \in [0, 1]$, any low-rank or sparse estimator for $M$ with $(n^2)$ parameters has $\Omega(\gamma^{-1})$ times the error of the sparse + low-rank estimator with $O(\gamma^{-1} n^{1.5} \log n)$ parameters.

*Proof of Theorem 3.* For each $i \in [n]$, let $q_i$ denote the $i$-th row of $Q \in \mathbb{R}^{n \times d}$. Define $J \in \mathbb{R}^{n \times n}$ to be the all 1s matrix. Define $T = M - J - QQ^\top$. Therefore,

$$T_{i,j} = \begin{cases} e - 2 & \text{if } i = j \\ e^{q_i^\top q_j} - 1 - q_i^\top q_j & \text{otherwise} \end{cases}.$$

By Hoeffding's inequality, for a pair $i \neq j$, we have that

$$\mathbb{P}\left(\left|q_i^\top q_j - \mathbb{E}[q_i^\top q_j]\right| \geq \epsilon\right) \leq 2 \exp\left(-\frac{2\epsilon^2}{\left(\frac{1}{\sqrt{d}} - \frac{-1}{\sqrt{d}}\right)^2}\right) = 2\exp(-d\epsilon^2/2).$$

Note that $\mathbb{E}[q_i^\top q_j] = 0$.

By a union bound over all pairs $i \neq j$ (there are $n(n-1)/2$ such pairs), with probability at least $1 - n^2 \exp\left(-d\epsilon^2/2\right)$, we have that

$$q_i^\top q_j \in [-\epsilon, \epsilon] \quad \text{for all } i \neq j.$$

Since we assume that $d \geq 6\epsilon^{-2} \log n$, we have that

$$n^2 \exp(-d\epsilon^2/2) \leq n^2 \exp(-3 \log n) = n^{-1}.$$

Hence $q_i^\top q_j \in [-\epsilon, \epsilon]$ for all $i \neq j$ with probability at least $1 - n^{-1}$. For the rest of the proof, we only consider this case (where $q_i^\top q_j \in [-\epsilon, \epsilon]$ for all $i \neq j$).

Since $1 + x \leq e^x \leq 1 + x + x^2$ for $|x| < 1$, we can bound the off diagonal elements $|T_{i,j}| \leq \epsilon^2$. In particular, for all $i \neq j$,

$$|T_{ij}| = \left| e^{q_i^\top q_j} - 1 - q_i^\top q_j \right| \leq \left( q_i^\top q_j \right) \leq \epsilon^2. \tag{4}$$

**Sparse + low-rank estimator:**  We use the following sparse + low-rank estimator:

$$E_{\mathrm{SL}} = \underbrace{(e-2) \cdot I}_{\text{sparse}} + \underbrace{J + QQ^\top}_{\text{low-rank}},$$

where $(e-2)I$ has row sparsity 1 and $\mathrm{rank}(J + QQ^\top) \leq d + 1 = O\left(\epsilon^{-2} \log n\right)$.

Notice that the $E_{\mathrm{SL}}$ estimate matches $M$ exactly on the diagonal, and on the off-diagonal it differs from $M$ by $T_{ij}$. Thus, the Frobenious error of the sparse + low-rank estimator is

$$\|M - E_{\mathrm{SL}}\|_F \leq \epsilon^2 \sqrt{n(n-1)} \leq \epsilon^2 n.$$

Set $\epsilon = \frac{\sqrt{\gamma}}{n^{1/4}}$ for $0 \leq \gamma \leq 1$, Then

(i) The sparse + low-rank parameter count is $n + n \cdot \mathrm{rank} \leq n \cdot O(\epsilon^{-2} \log n) \leq O(\gamma^{-1} n^{1.5} \log n)$.

(ii) The Frobenius error is $\leq \gamma \sqrt{n}$.

**Low-rank estimator:**  We want to argue that low-rank approximation would require more parameters. If we approximate the matrix $(e-2)I$ by a matrix $R$ with rank $r$, then the difference matrix will have at least $n - d$ singular values of magnitude $e - 2 \geq 1/2$. As a result, by the Eckart–Young–Mirsky theorem,

$$\|(e-2) \cdot I - R\|_F \geq \frac{1}{2} \sqrt{n-r}.$$

Define $T' = T - (e-2) \cdot I$, then $T'$ is all 0 on the diagonal and has absolute value $\leq \epsilon^2$ on off-diagonal entries. Thus $\|T'\|_F \leq \epsilon^2 n = \gamma \sqrt{n}$.

We want to show that if $R'$ is a rank $r'$ matrix, then $\|M - R'\|_F \geq \frac{1}{2} \sqrt{n - r' - d - 1} - \|T'\|_F$. We argue by contradiction. Suppose that there exists some matrix $R'$ with rank $r'$ such that

$$\|M - R'\|_F \leq \frac{1}{2} \sqrt{n - r' - d - 1} - \|T'\|_F.$$

Define $R = R' - J - QQ^\top$, so $M - R' = (e-2) \cdot I - R + T'$. We see that:

$$\begin{aligned}
\|(e-2) \cdot I - R\|_F &= \|M - R' - T'\|_F \\
&\leq \|M - R'\|_F + \|T'\|_F \\
&\leq \frac{1}{2} \sqrt{n - r' - d - 1} \\
&\leq \frac{1}{2} \sqrt{n - \mathrm{rank}(R)}.
\end{aligned}$$

This contradicts the result above, which states that $\|(e-2) \cdot I - R\|_F \geq \frac{1}{2} \sqrt{n - \mathrm{rank}(R)}$.

Therefore any low-rank estimator with rank $r$ such that $n - r = \Omega(n)$, which has $\Omega(n^2)$ parameters, will have error at least $\Omega(\sqrt{n - r - d - 1}) - \|T'\|_F = \Omega(\sqrt{n})$, which is $\Omega(\gamma^{-1})$ times the error of the sparse + low-rank estimator above.

**Sparse estimator:** For our sparse estimator, it is easy to see that for any $E_S \in \mathbb{R}^{n \times n}$ that has row sparsity $k$ (each row has fewer than $k$ non-zeros),

$$\|M - E_S\|_F \geq \Omega(\sqrt{n(n-k)}).$$

This implies that in order to achieve error $O(\sqrt{n})$, we would need $n - k = O(1)$, which requires $\Omega(n^2)$ parameters.

$\square$

Now we construct a matrix that shows better separation between the approximation capability of sparse + low-rank vs sparse or low-rank alone.

**Example 2.** *Consider the following randomized construction of matrix $Q \in \mathbb{R}^{n \times d}$ with $d \geq 6\epsilon^{-2}r \log n$ and $d = \Theta(\epsilon^{-2}r \log n)$ ($\epsilon \in (0, 1]$ and close to 0 and $r$ is $\Theta(\log n)$): each entry of $Q$ is picked independently and uniformly at random from $\{\pm\sqrt{r/d}\}$. Let $M = \sigma(QQ^\top)$ where $\sigma$ is the elementwise exponential function.*

Similar to Example 1, with high probability, we have:

$$(QQ^\top)_{i,j} = \begin{cases} r, & \text{if } i = j; \\ \in [-\epsilon, \epsilon], & \text{otherwise.} \end{cases}$$

We also have:

$$M_{i,j} = \begin{cases} e^r, & \text{if } i = j; \\ \in [1 - O(\epsilon), 1 + O(\epsilon)], & \text{otherwise.} \end{cases}$$

By setting $r$ appropriately, we can formalize the separation between the approximation ability of sparse, low-rank, and sparse + low-rank matrices:

**Theorem 4.** *Let $M$ be the attention matrix from Example 2. Any sparse or low-rank estimator of $M$ needs $\Omega(n^2)$ parameters for $O(n)$ error with probability at least $1 - n^{-1}$ while a sparse + low-rank estimator needs $O(n)$ parameters for $O(n)$ error with probability at least $1 - n^{-1}$.*

*Proof of Theorem 4.* Similar to the proof of Theorem 3, by Hoeffding's inequality, for a pair $i \neq j$, we have that

$$\mathbb{P}\left(\left|q_i^\top q_j - \mathbb{E}[q_i^\top q_j]\right| \geq \epsilon\right) \leq 2\exp\left(-\frac{2\epsilon^2}{\left(\frac{r}{\sqrt{d}} - \frac{-r}{\sqrt{d}}\right)^2}\right) = 2\exp\left(-\frac{d\epsilon^2}{2r}\right).$$

Note that $\mathbb{E}[q_i^\top q_j] = 0$. By a union bound over all pairs $i \neq j$ (there are $n(n-1)/2$ such pairs), with probability at least $1 - n^{-1}$ (since $d \geq 6\epsilon^{-2}r \log n$), we have that

$$q_i^\top q_j \in [-\epsilon, \epsilon] \quad \text{for all } i \neq j.$$

Since we assume that $d \geq 6\epsilon^{-2} \log n$, we have that For the rest of the proof, we only consider this case (where $q_i^\top q_j \in [-\epsilon, \epsilon]$ for all $i \neq j$).

Let $T = M - (e^r - 1) \cdot I + J$, where $J$ is the all one matrix. We see that $T$ is zero on the diagonal. Moreover, using the fact that $e^x \leq 1 + 2|x|$ for all $x \in [-1, 1]$, the off-diagonal entries of $T$ have of magnitude at most $2\epsilon$.

We consider 3 different estimators.

**Sparse + low-rank estimator:** Our estimator is

$$E_{\text{SL}} = \underbrace{(e^r - 1) \cdot I}_{\text{sparse}} + \underbrace{J}_{\text{low-rank}},$$

where $(e - 1)I$ has row sparsity 1 and $\text{rank}(J) = 1$.

The Frobenious error of sparse + low-rank approximation is

$$\|M - E_{\text{SL}}\|_F \leq O(\sqrt{\epsilon^2 n(n-1)}) \leq O(\epsilon n).$$

We have that:

(i) Sparse + low-rank parameter count is $n \cdot (1+1) \leq O(n)$.

(ii) Its Frobenius error is $\leq O(n)$.

**Low-rank estimator:** We want to argue that low-rank approximation would require more parameters. From a similar observation that any matrix $R$ with rank that $n - \text{rank} = \Omega(1)$,

$$\|(e^r - 1)I - R\|_F \geq \Omega(e^r),$$

(by Eckart–Young–Mirsky theorem), we obtain a similar result to the proof of Theorem 3.

If $R'$ is a matrix with rank such that $n - \text{rank} = \Omega(1)$, then $\|M - R'\|_F \geq \Omega(n) - \|T\|_F \geq \Omega(n) - O(\epsilon n) \geq \Omega(n)$. Hence any low-rank matrix with $O(n^2)$ parameters would have error $\Omega(n)$.

**Sparse estimator:** Similar to the proof of Theorem 3, for our sparse estimator, it is easy to see that for any $E_S \in \mathbb{R}^{n \times n}$ that has row sparsity $k$ (each row has fewer than $k$ non-zeros),

$$\|M - E_S\|_F \geq \Omega(\sqrt{n(n-k)}).$$

This implies that to get $O(n)$ error, we would need $\Omega(n^2)$ parameters. $\qquad\square$

## D.2 Generative Model, Softmax Temperature, and Matrix Approximation

Here we show 3 cases where depending on the softmax temperature, either we'll need low-rank, low-rank + sparse, or sparse to approximate the attention matrix.

We start with some notation first. Given a matrix $B$, let $B[i, j]$ be the entry in the $i$th row and $j$th column. For a range $[l, r]$, we define a matrix $B_{[l,r]}$ where $B_{[l,r]}[i, j] = B[i, j]$ if $B[i, j] \in [l, r]$ and $B_{[l,r]} = 0$ otherwise (that is, $B_{[l,r]}$ only keep entries for $B$ that are in the range $[l, r]$, with other entries zeroed out). We write $\text{supp}(C)$ for the set of locations of non-zeros in $C$. We let $\lambda_i(D)$ be the $i$-th largest (in absolute value) eigenvalue of $D$.

To prove Theorem 1, we first define a more general matrix class, prove that the attention matrix in Process 1 is a subset of this class (with high probability), and then show that Theorem 1 holds for this more general class. We introduce an extra parameter $l \in \mathbb{R}$, in addition to the inverse temperature $\beta$ and the intro-cluster distance $\Delta$.

**Matrix Class 1.** *Let $Q \in \mathbb{R}^{n \times d}$ with every row of $Q$ having $\ell_2$-norm in $[1 - O(\Delta), 1 + O(\Delta)]$, and let $A = QQ^\top$. Further:*

1. *Let $H = A_{[1/l, 2-1/l]}$ for some $l \geq \Omega(1)$. Assume that $H$ is block diagonal with $\Omega(n)$ blocks, and $\text{supp}(H)$ is $o(n^2)$. That is, the large entries of $QQ^\top$ form a block diagonal matrix.*
2. *Let $L = A - H$ then $L = A_{[-\Delta, \Delta]}$ where $\Delta = o(1/\log d)$. Assume that there is a constant fraction of elements in $\text{supp}(L)$ falling in $[0, \Delta]$. Assume that $\text{supp}(A_{[0,\Delta]})$ is $\Omega(n^2)$.*

*Let $M_\beta = \exp(\beta \cdot A)$.*

We now show that Process 1 is a subset of Matrix Class 1, with high probability.

**Lemma 5.** *The matrix $M_\beta$ in Process 1 is a subset of Matrix Class 1, where $l = \frac{1}{1-\Delta^2}$.*

*Proof.* We first bound the norm of each row in $Q$ in Process 1. For any $i, j$, we have

$$\|z_{ij}\|^2 = \|c_i + r_{ij}\|^2 = \|c_i\|^2 + 2c_i^\top r_{ij} + \|r_{ij}\|^2.$$

Since $c_i \sim \mathcal{N}(0, I_d/\sqrt{d})$, $\|c_i\|^2 \in [1 - \Delta^2, 1 + \Delta^2]$ with probability at least $1 - 2e^{-d\Delta^2/8}$ (by the standard argument using the fact that $\chi^2$-random variables are sub-exponential). Similarly, $\|r_{ij}\|^2 \in [\Delta^2 - \Delta^4, \Delta^2 + \Delta^4]$ with probability at least $1 - 2e^{-d\Delta^2/8}$. By concentration of measure, we can also bound $2c_i^\top r_{ij} \in [2\Delta - 2\Delta^3, 2\Delta + 2\Delta^3]$ as well. Therefore, we have that $\|z_{ij}\|^2 \in [1 - O(\Delta), 1 + O(\Delta)]$.

Now we show that the large entries of $QQ^\top$ form a block diagonal matrix. With high probability, the large entries come from intra-cluster dot product, and the small entries come from inter-cluster dot product.

We bound the intra-cluster dot product:

$$z_{ij}^\top z_{ik} = (c_i + r_{ij})^\top (c_i + r_{ik})$$
$$= \|c_i\|^2 + c_i^\top r_{ij} + c_i^\top r_{ik} + r_{ij}^\top r_{ik}.$$

Similar to the argument above, by concentration of measure, $\|c_i\|^2 \in [1 + \epsilon\Delta, 1 - \epsilon\Delta]$ with high probability (we will pick $\epsilon = \theta(\Delta)$). The cross terms $c_i^\top r_{ij}$ and $c_i^\top r_{ik}$ can be bounded using Cauchy-Schwarz inequality to be in $[-\epsilon\Delta, \epsilon\Delta]$ with high probability. And the fourth term $r_{ij}^\top r_{ik}$ is in $[-\epsilon\Delta^2, \epsilon\Delta^2]$ with high probability. Therefore, the inner product is in $1 \pm O(\epsilon\Delta)$ with high probability. This satisfies the first condition in Matrix Class 1, for $l = \frac{1}{1-\Delta^2}$, assuming $\epsilon \leq \Delta$.

We use a similar argument to bound the inter-cluster dot product. For $i \neq i'$

$$z_{ij}^\top z_{i'k} = (c_i + r_{ij})^\top (c_{i'} + r_{i'k})$$
$$= c_i^\top c_{i'}^\top + c_i^\top r_{i'k} + c_{i'}^\top r_{ij} + r_{ij}^\top r_{i'k}.$$

By concentration of measure, $c_i^\top c_{i'} \in [-\epsilon, \epsilon]$. Similar to the argument in the intra-cluster case, we can bound the other three terms, so this dot product is in $[-O(\epsilon), O(\epsilon)]$. This satisfies the second condition in Matrix Class 1.

$\square$

To prove Theorem 1 for Matrix Class 1, we start with some technical lemmas.

**Lemma 6.** *Let $F \in \mathbb{R}_{\geq 0}^{N \times N}$ be a symmetric matrix. Let $\lambda_{\max}$ be the largest eigenvalue of $F$. Assuming $N \geq 2$, we have that*
$$\lambda_{\max} \geq \min_{i \neq j} F[i,j].$$

*Proof.* Since $F$ is symmetric, $\lambda_{\max}$ is real and

$$\lambda_{\max} = \max_{u \neq 0} \frac{u^\top F u}{u^T u}.$$

Let $u$ be the all 1's vector, then

$$\lambda_{\max} \geq \frac{1}{N} \sum_{i=j} F[i,j]$$
$$\geq \frac{1}{N} \sum_{i \neq j} F[i,j]$$
$$\geq \frac{1}{N} \cdot N(N-1) \min_{i \neq j} F[i,j]$$
$$\geq \min_{i \neq j} F[i,j],$$

where the second step follows from all the diagonal entries are non-negative, the last step follows from $N \geq 2$ $\square$

The above implies the following result:

**Corollary 7.** *Let $F \in \mathbb{R}_{\geq 0}^{N \times N}$ be a block diagonal matrix. Let $r$ be the number of $m \times m$ blocks in $F$ for some $m \geq 2$. The $\lambda_r(F)$ is at least the smallest non-diagonal element in any $m \times m$ block ($m \geq 2$) in $F$.*

*Proof.* By Lemma 6, each $m \times m$ block $B$ ($m \geq 2$) by itself has max eigenvalue at least $\min_{i \neq j \in [m]} B[i,j]$. The claim then follows from the fact that any eigenvalue of $B$ is also an eigenvalue of $F$. $\square$

We'll need the following function for our low-rank argument:

$$f_k(x) = \sum_{i=0}^{k} \frac{x^i}{i!}.$$

Note that $f_\infty(x) = e^x$.

**Definition 1.** *Let $\epsilon \in (0, 1/10)$ and $L > 0$. We say a function $f : \mathbb{R} \to \mathbb{R}$ is $(\epsilon, L)$-close to $e^y$ if*

$$|e^y - f(y)| \le \epsilon \quad \text{for any } y \in [-L, L].$$

**Lemma 8.** *For any $\epsilon \in (0, 1/10)$ and $L > 0$. If*

$$D \ge 10(L + \log(1/\epsilon))$$

*then function $f_D(y)$ is $(\epsilon, L)$-close to $e^y$.*

*Proof.* Recall the definition of function $f_D$,

$$e^x = f_D(x) + \sum_{i=D+1}^{\infty} \frac{x^i}{i!},$$

It is sufficient to show that $|e^y - f(y)| < \epsilon$ if we have

$$\frac{x^{D+1}}{(D+1)!} \le \frac{\epsilon}{2},$$

We can show that

$$
\begin{aligned}
\frac{y^D}{D!} &\le \frac{L^D}{D!} \\
&\le \frac{L^D}{(D/4)^D} \\
&= (\frac{4L}{D})^D \\
&\le (1/2)^D \\
&\le \epsilon/10
\end{aligned}
$$

where the first step follows from $|y| \le L$, the second step follows $n! \ge (n/4)^n$, the forth step follows from $D \ge 10L$, the last step follows $D \ge 10\log(1/\epsilon)$ and $\epsilon \in (0, 1/10)$.

$\square$

We'll also use the following fact:

**Lemma 9.** *For any $D = o(\log n / \log d)$, we have*

$$\mathrm{rank}(f_D) \le n^{o(1)}.$$

*Proof.* We can upper bound $\mathrm{rank}(f_D(A))$ in the following sense:

$$
\begin{aligned}
\mathrm{rank}(f_D(A)) &\le (\mathrm{rank}(A))^D \\
&\le d^D \\
&= 2^{D \cdot \log d} \\
&= 2^{o(\log n)} \\
&= n^{o(1)}.
\end{aligned}
$$

where the second step follows from $\mathrm{rank}(A) \le d$, the forth step follows from $D = o(\frac{\log n}{\log d})$. $\square$

Finally we're ready to prove the theorem:

*Proof.* The basic idea is: (i) Use $f_{k^*}(b \cdot A)$ to get the low-rank approximation (ii) Use $\exp(b \cdot H)$ to get the sparse part.

**Small $\beta$ range,** i.e., $\beta$ is $o\left(\frac{\log n}{\log d}\right)$.

Low rank approximation: $R = f_{k^*}(b \cdot A)$.

Since each entry of $A$ is in $[-1, 1]$, each entry of $\beta \cdot A$ is in $[-\beta, \beta]$. But note that $\beta$ in this case is $o\left(\frac{\log n}{d}\right) = O(\log n \cdot \Delta)$. By the definition of $k^*$, each entry of $\exp(\beta \cdot A) - f_{k^*}(\beta \cdot A)$ has absolute value $\leq \epsilon$. Therefore the overall error is $\leq \epsilon n$.

For sparse only: By assumption, $m = \Omega(\|L\|_0)$ entries in $A$ are $\geq 0$, which are exactly the entries in $\exp(\beta \cdot A)$ that are $\geq 1$. Hence any (say) $\frac{m}{2}$ sparse approximation has error $\geq \sqrt{\frac{m}{2}} \geq \Omega(\sqrt{\|L\|_0})$. By our assumption, $\|L\|_0 = \Omega(n^2)$.

**Mid-range $\beta$,** i.e., $\beta \geq \frac{1}{l} \cdot \log n$ and $\beta$ is $O(\log n)$.

Sparse only: the argument is the same as in the low $\beta$ range.

Sparse + low-rank: The low-rank part $R = fst(\beta \cdot A)$. By Lemma 9, this has rank $n^{o(1)}$, so it has $n^{(1+o(1))}$ parameters.

The sparse part is $S = e^{\beta \cdot H} - R_{\text{supp}(H)}$. Clearly this needs $|\text{supp}(H)|$ parameters.

Let $E = M_\beta - (S+R)$. Then (i) in $\text{supp}(H)$, $E$ is all 0. (ii) output of $\text{supp}(H)$, by definition, entries of $\beta \cdot A$ are in $[-\beta\Delta, \beta\Delta]$, which in the current range of $\beta$ is $[-O(\log n\Delta), O(\log n\Delta)]$. Therefore all the entries of $E$ have absolute value $\leq \epsilon$. By the definition of $k^*$, we have that $\|E\|_F \leq \epsilon n$.

Low-rank only: Let $\widetilde{R}$ be rank $r - n^{o(1)} - 1$ that approximates $M_\beta$. Then using the same argument as our existing lower bound argument, we get that $\widetilde{R} - R \approx_E S$ (this means that the error $\leq \|E\|_F + \left\|M_\beta - \widetilde{R}\right\|_F$). Now note that $S = e^{\beta \cdot H} - (f_{k^*}(\beta \cdot A))_{\text{supp}H}$ is a symmetric, block diagonal matrix with $r = \Omega(n)$ blocks. Corollary 7 implies that $\lambda_r(S)$ is at least the smallest non-diagonal value in $S$. Now the smallest non-diagonal value in $e^{\beta \cdot H}$ is $\geq e^{\frac{1}{l}\log n} = n$. On the other hand, the largest value in $(f_{k^*}(\beta \cdot A))_{\text{supp}H}$ is

$$\leq k^* \frac{\beta^{k^*}}{k^*!} \leq \beta \cdot \left(\frac{e\beta}{k^* - 1}\right)^{k^* - 1}$$
$$\lesssim \log n \left(\frac{e \cdot \log n}{\log n \cdot \Delta}\right)^{O(\log n \cdot \Delta)}$$
$$\lesssim \log n e^{O(\log n \cdot \Delta \cdot \log \frac{1}{\Delta})}$$
$$\lesssim \log n \cdot n^{o(1)}$$
$$= n^{o(1)}.$$

Hence $\lambda_r(S)$ is $\Omega(n)$. The claimed result then follows since $\|E\|_F \leq \epsilon n$ and $\text{rank}\widetilde{R} - R \leq r - 1$ (Eckart-Young-Mirsky theorem).

**Large $\beta$ range,** i.e., $\beta \geq \omega(\log n)$.

Sparse only: $S = e^{\beta \cdot H}$. Note that each entry in $E = M_\beta - S$ is upper bounded by $e^{\Delta \cdot \beta} \leq e^{o\left(\frac{\beta}{\log d}\right)}$. Then

$$\|E\|_F \leq n \cdot e^{o\left(\frac{\beta}{\log d}\right)}$$
$$\leq \epsilon \cdot e^{\log \frac{n}{\epsilon} + o\left(\frac{\beta}{\log d}\right)}$$
$$\leq \epsilon \cdot e^{o(\beta) + o\left(\frac{\beta}{\log d}\right)}$$
$$\leq \epsilon \cdot e^{o(\beta)}$$
$$\leq \epsilon \cdot e^{\beta/l}.$$

Low-rank only: since $\|E\|_F$ is $\leq \epsilon e^{\beta/l}$, it is enough to argue that any rank $r$-approximation to $S$ has error $\geq e^{\beta/l}$. But the latter follows since $\lambda_r(S) \geq e^{\beta/l}$. This is because $e^{b \cdot H}$ is symmetric and

each entry in $H$ is $\geq \frac{1}{\lambda}$. Then we can use Corollary 7. Eckart-Young-Mirsky then completes the proof. $\qquad\square$

## D.3 Scatterbrain: Analysis

Here we prove Theorem 2, which shows that Scatterbrain approximation is unbiased and analyses its variance. We restate the theorem here for the reader's convenience.

**Theorem.** *Define* $\sigma(q,k) = \exp(q^\top k)$, $\widehat{\sigma}^{\mathsf{pfe}}$ *as Performer's estimator and* $\widehat{\sigma}^{\mathsf{sbe}}$ *as Scatterbrain estimator. Denote* $\mathcal{S}^{d-1} \subset \mathbb{R}^d$ *as the unit sphere. Suppose* $q,k \in \mathcal{S}^{d-1}$ *are such that* $\|q - k\| < \tau$. *Then:*

$$\mathbb{E}[\widehat{\sigma}^{\mathsf{sbe}}(q,k)] = \sigma(q,k), \quad \mathrm{Var}[\widehat{\sigma}^{\mathsf{sbe}}(q,k)] = (1-p) \cdot \mathrm{Var}[\widehat{\sigma}^{\mathsf{pfe}}(q,k)] < \mathrm{Var}[\widehat{\sigma}^{\mathsf{pfe}}(q,k)]$$

*where* $p = \exp(-\frac{\tau^2}{4-\tau^2}\ln d - O_\tau(\ln\ln d))$.

*Proof.* Let $A_{ij} = \exp(q_k^\top k_j)$ be $ij$-entry of the unnormalized attention matrix, $A_{ij}^{\mathrm{lr}} = \phi(q_i)^\top \phi(k_j)$ the entry of the low-rank approximation (Performer), and let $A_{ij}^{\mathrm{sb}}$ be the entry of the Scatterbrain (sparse + low-rank) approximation. By the construction of the Scatterbrain attention matrix (Eq. (1)), if $ij \in \mathcal{S}$, where $\mathcal{S}$ is the set of indices selected by the LSH, then:

$$A_{ij}^{\mathrm{sb}} = (\widetilde{Q}\widetilde{K}^\top + S)_{ij} = \phi(q_i)^\top \phi(k_j) + \exp(q_i^\top k_j) - \phi(q_i)^\top \phi(k_j) = \exp(q_i^\top k_j).$$

If $ij \notin \mathcal{S}$, then

$$A_{ij}^{\mathrm{sb}} = (\widetilde{Q}\widetilde{K}^\top + S)_{ij} = \phi(q_i)^\top \phi(k_j) + 0 = \phi(q_i)^\top \phi(k_j).$$

In other words, $A^{\mathrm{sb}}$ matches $A$ on the indices in $\mathcal{S}$, and matches $A^{\mathrm{lr}}$ on the indices not in $\mathcal{S}$.

To show that $A^{\mathrm{sb}}$ is an unbiased estimator of $A$, we simply use the fact that $A^{\mathrm{lr}}$ is also an unbiased estimator of $A$ [16, Lemma 1]:

$$\begin{aligned}
\mathbb{E}[A_{ij}^{\mathrm{sb}}] &= \mathbb{P}(ij \in \mathcal{S})\mathbb{E}[A_{ij} \mid ij \in \mathcal{S}] + \mathbb{P}(ij \notin \mathcal{S})\mathbb{E}[A_{ij}^{\mathrm{lr}} \mid ij \notin \mathcal{S}] \\
&= \mathbb{P}(ij \in \mathcal{S})A_{ij} + \mathbb{P}(ij \notin \mathcal{S})A_{ij} \\
&= A_{ij}.
\end{aligned}$$

In other words, $\mathbb{E}[\widehat{\sigma}^{\mathsf{sbe}}(q,k)] = \sigma(q,k)$.

Now we analyze the per-entry variance of $A^{\mathrm{sb}}$. Since $A^{\mathrm{sb}}$ is an unbiased estimator of $A$, by the law of total variance,

$$\begin{aligned}
\mathbb{Var}(A_{ij}^{\mathrm{sb}}) &= \mathbb{P}(ij \in \mathcal{S})\mathbb{Var}(A_{ij} \mid ij \in \mathcal{S}) + \mathbb{P}(ij \notin \mathcal{S})\mathbb{Var}(A_{ij}^{\mathrm{lr}} \mid ij \notin \mathcal{S}) \\
&= \mathbb{P}(ij \in \mathcal{S}) \cdot 0 + \mathbb{P}(ij \notin \mathcal{S})\mathbb{Var}(A_{ij}^{\mathrm{lr}}) \\
&= \mathbb{P}(ij \notin \mathcal{S})\mathbb{Var}(A_{ij}^{\mathrm{lr}}).
\end{aligned}$$

To compute the probability that the index $ij$ is not in $\mathcal{S}$ (i.e., not selected by LSH), we use the standard bound on cross-polytope LSH [3, Theorem 1]:

$$p := \mathbb{P}(ij \in \mathcal{S}) = \exp(-\frac{\tau^2}{4 - \tau^2}\ln d - O_\tau(\ln\ln d)).$$

Therefore,

$$\mathbb{Var}(A_{ij}^{\mathrm{sb}}) = (1-p)\mathbb{Var}(A_{ij}^{\mathrm{lr}}) < \mathbb{Var}(A_{ij}^{\mathrm{lr}}).$$

In other words, $\mathbb{Var}[\widehat{\sigma}^{\mathsf{sbe}}(q,k)] = (1-p) \cdot \mathbb{Var}[\widehat{\sigma}^{\mathsf{pfe}}(q,k)] < \mathbb{Var}[\widehat{\sigma}^{\mathsf{pfe}}(q,k)]$.

More explicitly, by plugging in the variance of $A^{\mathrm{lr}}$ [16, Lemma 2], we have

$$\mathbb{Var}(A_{ij}^{\mathrm{sb}}) = (1-p)\frac{1}{m}\exp\left(\|q_i + k_j\|^2\right)\exp(2q_i^\top k_j)\left(1 - \exp\left(-\|q_i + k_j\|^2\right)\right),$$

where $p = \exp(-\frac{\tau^2}{4-\tau^2}\ln d - O_\tau(\ln\ln d))$

$\qquad\square$

# E   Additional Experiments and Details

## E.1   Datasets

**ImageNet [24]:** ImageNet is one of the most widely-used image classification benchmarks. In our experiments in Section 5.1 of evaluating the approximation accuracy of Scatterbrain, both BigGAN and Vision Transformer are pre-trained on this dataset. It has roughly 1.2 million training images and 50,000 validation images.

**WikiText103 [42] and Copy [33]:** WikiText103 is a popular dataset for auto-regressive models. It is from a collection of over 100 million tokens extracted from the set of verified good and featured articles on Wikipedia. It has 28,475 training articles, 60 for validation and 60 for testing.

Copy is a synthetic a synthetic sequence duplication task where inputs are of the form $0w0w$ and $w \in \{0, ..., N\}^*$. It is previously used in [33, 14]. This task is useful for demonstrating the effectiveness of long range attention: it requires non-local attention lookups. It cannot be solved by any model relying on sparse attention with a limited range such as, local attention.

**Long Range Arena (LRA) [54]:** This is a recent benchmark for evaluating efficient transformers with long input sequence. We used ListOps [43], byte-level IMDb reviews text classification [41], byte-level document retrieval [45], image classification on sequences of pixels [34] and Pathfinder [37]. We follow the same evaluation mechanism from [54] but implement our own version in Pytorch (like data loader).

**GlUE [61]:** GLUE is a standard multi-task benchmark in NLP. It has single-sentence tasks, CoLA and SST-2; similarity and paraphrase tasks, MRPC, STS-B, QQP; and inference tasks, MNLI, QNLI, RTE and WNLI. For our additional experiments below (not enough space to be included in the main paper), we follow the tradition from [25, 65, 21] and truncate all the input sequences to 128 tokens.

## E.2   Settings

**BigGAN:** We adapt the same pre-trained BigGAN model from [21] with no additional training. The model has a single attention layer at resolution $64 \times 64$ (4096). Similar to the prior work, we also replace its full attention layer with Scatterbrain at the same resolution. Figure 5 in the main paper shows the best-effort comparison with [1/32, 1/16, 1/8, 1/4, 1/2] of the parameter budget. For example, if given parameter budget 1/2, we report the best performance of Smyrf from choice of 32/64/128 hash round 64/32/16 cluster size.

**T2-ViT:** We use the pre-trained vision transformer model T2T-ViT-14 from [66] with $224 \times 224$ image size. Without any additional training, we just replace the attention layer with Scatterbrain and other baselines and evaluate the approximation error and classification accuracy on ImageNet testings. Again, we report the best-effort best performance of each approximation given the certain parameter budget.

**Auto-regressive Model:**   We follow the settings from the popular repo `https://github.com/NVIDIA/DeepLearningExamples` for training vanilla Transformer from scratch on WikiText103, except for chunking WikiText103 into sequence length 1024 in order to simulate long input sequences. The model is 16 layer with 8 head and 512 model dimension. We train all the models for 30 epochs and report the best Testing Perplexity. The model we use for Copy task is simply a 2-layer-4-head transformer and sequence length is also 1024. We make 5 runs and report average. Table 4 presents the results with standard deviation.

**Classification Model:**   We follow the model setting from [54, 64]. We share the same finding with [64] that the acuracy for the Retrieval tasks is actually higher than reported in [54].

**Ratio between Sparse and Low-rank components:** There are some rules that we used in our experiments to set this ratio. For inference, we set this ratio based on the entropy of an observed subset of attention matrices in different layers: we allocate more memory to the low-rank component compared to the sparse component if the entropy is high. For training, generally allocating more memory budget to sparse tends to perform better, so in the experiment, we set the ratio to 3:1 (sparse: low-rank component) for simplicity. Moreover, in future work, it could be useful to make this ratio adaptive during training. For example, in the early stage of the training and early layers, attention matrices are usually more uniform (higher entropy). Thus, the approximation error could be

even lower if the ratio favors low-rank-based components. One approach could be to monitor the approximation error of sparse and low-rank components compared to full attention regularly and adjust the memory budget accordingly. We will add the above discussion to the updated manuscript.

Table 4: The performance of Scatterbrain, REFORMER, PERFORMER and Full-Attention on Long-Range-Arena benchmarks and 2 popular language modeling tasks. We fix the same number of parameters ($1/8$ of the full) used for approximating the attention matrix for each method.

| Attention | Copy (ppl) | WikiText-103 (ppl) | Attention | ListOps | Text | Retrieval | Image | Pathfinder | Avg |
|---|---|---|---|---|---|---|---|---|---|
| Full Attention | 1 | 25.258±0.37 | Full Attention | 38.2±0.17 | 63.29±0.38 | 80.85±0.12 | 41.78±0.26 | 73.98±0.31 | 59.62 |
| Reformer | 6.8±0.64 | 27.68±0.53 | Reformer | 36.85±0.37 | 58.12±0.42 | 78.36±0.29 | 28.3±0.39 | 67.95±0.28 | 53.9 |
| Performer | 49±2.7 | 66±5.8 | Performer | 35.75±0.29 | 62.36±0.49 | 78.83±0.33 | 39.71±0.48 | 68.6±0.36 | 57.05 |
| Scatterbrain | **2.58±0.21** | **26.72±0.44** | Scatterbrain | **38.6±0.22** | **64.55±0.34** | **80.22±0.31** | **43.65±0.46** | **69.91±0.25** | **59.38** |

## E.3 More Ablation Studies

### E.3.1 Memory Budget

We present an ablation study on the parameter budget for the WikiText-103 language modeling task. We show that Scatterbrain outperforms its sparse and low-rank baselines across a range of parameter budgets. The results are presented in Table 5.

**Analysis:** We have observed that Scatterbrain outperforms its sparse and low-rank baselines under different memory budgets. Similar to what we found in Section 5.2, Performer does not train stably even with $\frac{1}{4}$ of the full attention memory. However, under the Scatterbrain framework, Performer can be combined with Reformer in an elegant way to achieve the same accuracy while using only half of the memory and faster than Reformer by exploiting the sparse+low-rank structure in attention matrices.

Table 5: We run WikiText-103 LM with a sweep of 1/4, 1/8, 1/16 memory budget. We show the validation perplexity and speed-up with respect to the full attention with different efficient Attention layers.

| | $\frac{1}{4}$ Mem | $\frac{1}{8}$ Mem | $\frac{1}{16}$ Mem |
|---|---|---|---|
| | Perplexity (Speed-up) | Perplexity | Perplexity |
| SMYRF | 26.76 (1.6×) | 27.68 (1.39×) | 28.7(1.85×) |
| PERFORMER | 58(2.13×) | 66 (2.01×) | 85(1.77×) |
| Scatterbrain | **26.26(1.58×)** | **26.72 (1.87×)** | **27.74(2.03×)** |

### E.3.2 Different Sparse and Low-rank baselines

Scatterbrain is general enough to accommodate different kinds of sparse and low-rank approximations as its sub-components. In particular, we can combine Local attention or block sparse (from Sparse Transformer and BigBird) + Performer (instead of Reformer + Performer) in a similar fashion. The support of the sparse matrix S will thus be fixed and not adaptive to input, but all the other steps are exactly the same.

We have run additional experiments on the Local attention + Performer combination and BigBird. Recall that in Appendix E, we have shown Scatterbrain can reduce the attention memory of Vision Transformer by 98% at the cost of only 0.8% drop of accuracy when serving as a drop-in replacement for full attention without training on ImageNet. We show the results for local+performer variation with the same memory budget in Table 6.

We have also run additional experiments on Local attention on Copy and Wikitext-103 language modeling task ( Table 7). We see that Local attention is reasonably competitive on Wikitext-103 but does not perform well on Copy. The results are not surprising as noted in the Reformer paper that Copy requires non-local attention lookups.

Table 6: Top-1 Accuracy of pre-trained T2T Vision Transformer on ImageNet with different attention replacements. Error represents the average normalized approximation error to full attention.

| Attention | Top-1 Acc |
|---|---|
| Full Attention | 81.7% |
| SMYRF | 79.8% |
| Local | 79.6% |
| Performer | 80.1% |
| BigBird | 80.3% |
| Scatterbrain (Local + Performer) | 80.3% |
| Scatterbrain (SMYRF + Performer) | **80.7**% |

Table 7: Additional experiments for Local attention on the Copy and Wikitext-103 language modeling task.

| **Attention** | Copy (ppl) | WikiText-103 (ppl) |
|---|---|---|
| Full Attention | 1 | 25.258 |
| Reformer | 6.8 | 27.68 |
| Performer | 49 | 66 |
| Local | 53 | 30.72 |
| Scatterbrain | **2.58** | **26.72** |

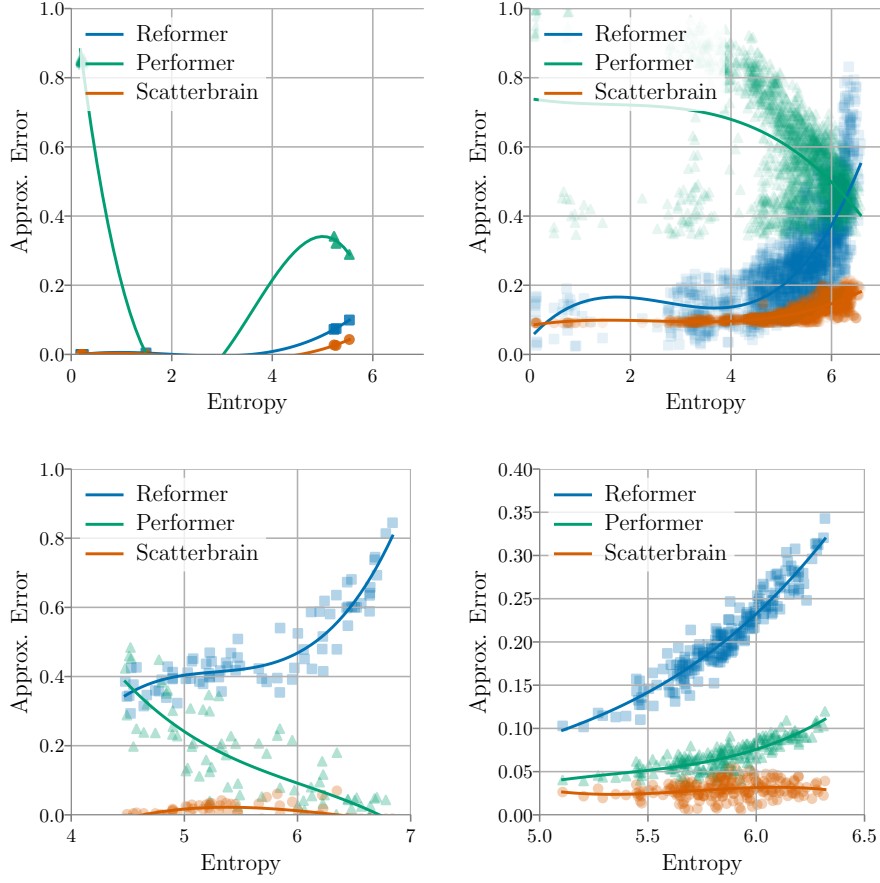

Figure 8: Top two plots present Approximation Error vs. Entropy of attention matrices for REFORMER, PERFORMER and Scatterbrain on Copy (left) and WikiText103 (right). Bottom two plots present Approximation Error vs. Entropy of attention matrices for REFORMER, PERFORMER and Scatterbrain on Text-IMDb (left) and Image-Cifar10 (right). Recall we observe that entropy of the softmax attention distribution (i.e., scale of logits) determines the regimes where sparse, low-rank, or sparse + low-rank perform well. Scatterbrain yields better approximation than REFORMER or PERFORMER in most of the cases; PERFORMER performs the worst on language modeling tasks while REFORMER performs the worst on classification tasks. These plots for approximation error analysis match with their performance on downstream tasks.

### E.3.3 Different Sparse and Low-rank baselines

### E.4 Analysis

Recall in Section 5, we have reported the analysis after visualizing the error of REFORMER (sparse), PERFORMER (low-rank), and Scatterbrain (sparse + low-rank) given the same number of parameters when approximating the full attention matrices for each attention layer during training. In Figure 8, we present the visualization.

The conclusion for language modeling tasks is that sparse+low-rank has the smallest approximation error in most of the cases, and sparse has the largest error, which matches with the end-to-end results. It also confirms the observation in the popular benchmark paper [54] that kernel or low-rank based approximations are less effective for hierarchical structured data. For classification tasks, we again find that Scatterbrain has the smallest approximation error, while PERFORMER is the worst on ListOps and REFORMER has the largest error on classification tasks, which matches with the end-to-end results and confirms our observations earlier (sparse and low-rank approximation excel in different regimes).

### E.5 Additional Experiments of Fine-tuning Bert on GLUE

We provide additional experiments of fine-tuning Bert on GLUE in Table 8. We follow the similar setting as [21]. We replace all the attention layers in Bert base model with Scatterbrain and other baselines. Then we fine-tune Bert on 9 downstream tasks for 3 epochs with batch size 32 and learning rate 3e-5. The parameter budget is 1/2 of the full attention because sequence length 128 is not very long. We can see Scatterbrain outperforms all the other baselines in most of the downstream tasks.

Table 8: Results of GLUE when replacing dense attention matrices with SMYRF, PERFORMER and Scatterbrain in BERT base model. We fix the same number ofparameters (1/2 of the full) used for approximating the attention matrix for each method.

|  | CoLA | SST-2 | MRPC | STS-B | QQP | MNLI | QNLI | RTE | WNLI |
|---|---|---|---|---|---|---|---|---|---|
|  | mcc | acc | acc | corr | acc | acc | acc | acc | acc |
| FULL | 0.576 | 0.934 | 0.874 | 0.879 | 0.905 | 0.813 | 0.916 | 0.668 | 0.43 |
| SMYRF | 0.538 | 0.912 | 0.833 | 0.856 | 0.898 | 0.775 | 0.879 | **0.626** | 0.412 |
| PERFORMER | 0.508 | 0.838 | 0.782 | 0.203 | 0.831 | 0.563 | 0.763 | 0.556 | **0.449** |
| Scatterbrain | **0.569** | **0.927** | **0.863** | **0.867** | **0.902** | **0.813** | **0.893** | 0.619 | 0.428 |

## F   Further Discussions and Future Work

In this paper, we present Scatterbrain, unifying the strength of sparse and low-rank approximation. It is inspired by the observations on the attention matrix structures induced by the data and softmax function as well as the classical robust-PCA algorithm. In our implementation and analysis, we have REFORMER/Smyrf and PERFORMER as the back-bone for sparse and low-rank approximations because of their properties, e.g. Performer is unbiased. Scatterbrain is fundamentally a framework for combining the strength of sparse and low-rank variants, so it can be easily extended to other variants, such as Routing Transformer [50] or Nystromformer [64]. Further more, our observations on the connection between entropy and low-rank/sparse approximation error also provide an opportunity for efficiently detecting the approximation or compression method to choose for different architectures or benchmarks.