# OpenReview forum: "Scatterbrain: Unifying Sparse and Low-rank Attention"
_NeurIPS.cc/2021/Conference — NeurIPS 2021 Poster_

### Official Review · Reviewer_Hndw · 2021-07-10

**Rating:** 8
**Confidence:** 3

**Summary:**

Recently, sparse- and low-rank approximations are proposed to reduce the computational cost of the Transformer. However, each approximation works well in different areas. The authors show that the softmax temperature (entropy) determines which approximation is more appropriate. To enjoy the best of both approaches, this paper proposes Scatterbrain, which uses both sparse- (as Reformer) and low-rank (as Performer) attention computation. The key idea is to make sparse matrix S to match (A – QK) instead of A. Experiments on both pre-trained networks and end-to-end training show that Scatterbrain can achieve much smaller approximation error and better performance, respectively.

**Limitations And Societal Impact:**

The authors well addressed the limitations and potential negative social impact.

**Main Review:**

Overall, the idea and the motivation are clear. The paper is well-written and easy to follow. The selection of baselines is appropriate, and they are properly compared with the proposed algorithm. The analysis of the attention temperature is novel and meaningful. The result is also well visualized (ex: Fig 4, 5). Using robust-PCA as the most accurate approximation also helps readers to get the sense of the ‘golden’ combination.

Suggestions:

(1)	I think figure 2 needs more explanations. (… what points and color mean, how they are visualized, etc.) In fact, is the figure necessary?

(2)	It would be great if a simpler way (just add S + QK, line (198-200)) is also compared in experiments.

(3)	In figure 6, the legend should be Reformer -> SMYRF.


**Time Spent Reviewing:**

5

---

> ### Author Response · Authors · 2021-08-10
> **Reply to Review by Reviewer Hndw**
>
> We appreciate your concise and precise summarization of our work! We are glad that you find our analysis of the attention temperature **novel** and **meaningful**.
>
> **Q1: I think figure 2 needs more explanations. (… what points and color mean, how they are visualized, etc.) In fact, is the figure necessary?**
>
> **R:** The figure visualizes the generative process of clustered elements in the input sequence, which provides an intuition on how the sparse + low-rank structure in attention matrices could arise. Specifically, each point represents a vector z_{ij} in Process 1 (line 134), and each color represents the cluster identity (1, …, C in Process 1). We will add clarification to the figure caption in the updated manuscript.
>
> **Q2: It would be great if a simpler way (just add S + QK, line (198-200)) is also compared in experiments.**
>
> **R:** Thanks for your great suggestion! We have added additional experiments with S + QK.
> We see that S+QK indeed results in larger approximation error and poor downstream task performance, which empirically verifies the necessity of an exact algorithm.
>
> --------------------------
> Experimental Details: Recall that in Section 5.1, we have shown Scatterbrain can reduce the attention memory of Vision Transformer by 98% at the cost of only 0.8% drop of accuracy when serving as a drop-in replacement for full attention without training on ImageNet. We show the results for S+QK with the same memory budget in the following table.
>
> | Attention | Top-1 Acc | Error (avg) |
> | ----------- | ----------- | ----------- |
> | Full Attention | 81.7% | - |
> | SMYRF |  79.8%  | 11.4% |
> | Performer |  80.1%  | 7.5% |
> | S+QK |  79.7%  | 12.6% |
> | Scatterbrain |  80.7%  | 5.3% |
>
> Analysis: We see that S+QK results in less accurate approximation (2.5x approx. error and 1 point drop in accuracy) due to double counting especially during inference.
>
> **Q3: In figure 6, the legend should be Reformer -> SMYRF.**
>
> **R:** Thanks for pointing it out! We have fixed this typo.

---

### Official Review · Reviewer_QK3S · 2021-07-16

**Rating:** 7
**Confidence:** 4

**Summary:**

The work proposes to combine sparse attention and low-rank attention to form a better approximation to full attention. Theoretical analysis shows sparse attention is particularly good for approximating low-entropy attention while low-rank attention is good for the high-entropy case. By combining the two, the proposed Scatterbrain is able to outperform both low-rank and sparse attention alone on post-training approximation with BigGAN and T2T-ViT as well as end-to-end training on language modeling and long range arena. As a result, Scatterbrain makes the gap between full attention and linear-complexity attention closer.

**Limitations And Societal Impact:**

The limitations are clearly discussed in the paper and I don't see any potential negative societal impact.

**Main Review:**

The idea of combining sparse attention and low-rank attention is intuitively reasonable and well motivated by the theoretical analysis. Although there is still a clear gap compared to the full attention, the empirical results is already encouraging.

One suggestion is to add some study of the impact of parameter budget under language modeling (as it's more realistic) as well as the speed & memory test (Fig 6). This will provide additional helpful insights, especially for practical applications.

**Time Spent Reviewing:**

3

---

> ### Author Response · Authors · 2021-08-10
> **Reply to Review by Reviewer QK3S**
>
> Thanks for your encouragement in our theoretical analysis! We appreciate your suggestions on our experiments, which have helped us improve the paper.
>
> **Q: One suggestion is to add some study of the impact of parameter budget under language modeling (as it's more realistic) as well as the speed & memory test (Fig 6). This will provide additional helpful insights, especially for practical applications.**
>
> **R:** We have added an ablation study on the parameter budget for the WikiText-103 language modeling task. We show that Scatterbrain outperforms its sparse and low-rank baselines across a range of parameter budgets. The results are presented in the following table.
>
> -------------------------------
> Experimental Details: We run WikiText-103 LM with a sweep of 1/4, 1/8, 1/16 memory budget. We show the validation perplexity and speed-up with respect to the full attention with different efficient Attention layers.
>
> | Attention | 1/4 Mem | 1/8 Mem | 1/16 Mem |
> | ----------- | ----------- | ----------- | ----------- |
> |  | Perplexity (Speed-up) | Perplexity (Speed-up) | Perplexity (Speed-up) |
> | Reformer | 26.76(1.6x)| 27.68 (1.39x)| 28.7(1.85x)
> | Performer | 58(2.13x)| 66 (2.01x)| 85(1.77x)
> | Scatterbrain | 26.26(1.58x)| 26.72 (1.87x) | 27.74(2.03x)
> -------------
>
> Analysis: We have observed that Scatterbrain outperforms its sparse and low-rank baselines under different memory budgets. Similar to what we found in 5.2.1, Performer does not train stably even with 1/4 of the full attention memory. However, under the Scatterbrain framework, Performer can be combined with Reformer in an elegant way to achieve the same accuracy while using only half of the memory and faster than Reformer by exploiting the sparse+low-rank structure in attention matrices.

---

### Official Review · Reviewer_6aFS · 2021-07-16

**Rating:** 5
**Confidence:** 5

**Summary:**

This paper proposes ScatterBrain a combination of sparse and low-rank attention and shows that it outperforms both Reformer and Performer on language modeling, Long range arena and vision tasks.

**Limitations And Societal Impact:**

Yes, i have no concerns.

**Main Review:**

Essentially, despite what this paper is trying hard*not* to be, this seems like a simple combination of sparse (Reformer) LSH attention with low rank (Performer) attention. I would rate the novelty quite low especially with the way the methods are combined (just adding Reformer attention with Performer's low rank attention).

I think many prior work (BigBird, ETC, Luna, Long short Transformers etc) have already shown that a combination of multiple sparse patterns or global components are useful. Along this line, I think the narrative of multiple attention being useful is not entirely new either.

If this paper was pitching itself "evaluating the effectiveness of a combination of different xformer variants", I can see how this is useful but the number of examples enumerated here is only a single method. I would encourage the authors to reframe this paper as such and evaluate more components rather than pitching a single combination (Reformer + Performer). Still, it is great to know that this outperforms standalone Performer and Reformer but i think the contribution is not really enough.

Furthermore, I also have some concerns about experiments, namely:

1. The performance of Performer on Copy and Wikitext is horrible. Have the authors tuned the model carefully? I think the kernel choice in Performer is important, it would be good to know if Performer was adequately tuned.
2. For Long Range arena, Reformer and Performer are only reported. Given that there are other models evaluated in LRA, why did the authors not compare with the other models? For example, did the model outperform BigBird in the LRA paper?
3. For language modeling, can the authors provide a strong local attention baseline? I think local attention should always at least be reported whenever xformer comes into play, given that it is the "simple naive baseline"

More questions:
1. The sparse variant here is Reformer, which is form of learnable sparsity. What about fixed sparse patterns, like Sparse Transformer or Longformer/Sparse Transformer?

More thoughts and suggestions:
1. Try combining different sparse and low-rank models, and draw insights from that. Are there benefits of different composition of attention types? What about different types of sparse models.
2.  What about interleaving layers? Does interleaving Reformer and Performer give on the necessary inductive bias as ScatterBrain?


**Time Spent Reviewing:**

0.5

---

> ### Author Response · Authors · 2021-08-10
> **Reply to Review by Reviewer 6aFS**
>
> We thank the reviewer for the constructive and detailed suggestions! We have carefully thought through all your great questions and provide details below. We split our response into 3 posts: (1) contributions & novelty (2) additional experiments (3) other detailed questions.
>
> ***Contributions & Novelty***
>
> **Q1: Essentially, despite what this paper is trying hard not to be, this seems like a simple combination of sparse (Reformer) LSH attention with low rank (Performer) attention. I would rate the novelty quite low especially with the way the methods are combined (just adding Reformer attention with Performer's low-rank attention).**
>
> **R:** We emphasize that our goal is not to simply combine two kinds of attention (sparse and low-rank), but (i) to understand their strengths and weaknesses, and (ii) to show how sparse + low-rank (a fundamental matrix structure) can approximate attention matrix **strictly** better than sparse or low-rank alone. The sparse + low-rank matrix structure has been well studied in statistics and signal processing since the late 2000s / early 2010s [1]. This structure naturally generalizes low-rank [2, 3] and sparse [4] matrices. However, as the reviewer notes, despite the multitude of papers, this sparse + low-rank matrix approximation has not been rigorously studied in the context of attention matrices. We undertook this study in this manuscript, showing how we can relax the sparse + low-rank approximation from robust PCA (which is too slow and memory intensive for attention), making it efficient while still retaining PCA’s accuracy.
>
> Moreover, by formulating the problem as matrix approximation with sparse + low-rank, we design Scatterbrain to be **general**---“it can use different kinds of low-rank and sparse approximation as its sub-components'' (line 236)---and the combination of Reformer / Smyrf + Performer is simply one instantiation of Scatterbrain. Instead of using Reformer/Smyrf as a sparse component, we could use local attention or random block-sparse attention as the reviewer suggests. Instead of using Performer as a low-rank component, we could also use Linear attention [5] or global tokens as in BigBird. The Scatterbrain method would work exactly the same way. As long as the low-rank component is unbiased (e.g., Performer), its combination with any sparse component in Scatterbrain would yield an unbiased estimator of the attention matrix. We will add this analysis, and further clarify the generality of Scatterbrain in the updated manuscript.
>
> [1] Candès, E. J., Li, X., Ma, Y., & Wright, J. (2011). Robust principal component analysis?. Journal of the ACM (JACM), 58(3), 1-37.
>
> [2] Hotelling, H. (1933). Analysis of a complex of statistical variables into principal components. Journal of educational psychology, 24(6), 417.
>
> [3] Udell, M., & Townsend, A. (2019). Why are big data matrices approximately low rank?. SIAM Journal on Mathematics of Data Science, 1(1), 144-160.
>
> [4] Tewarson, R. P., & Tewarson, R. P. (1973). Sparse matrices (Vol. 69). New York: Academic Press.
>
> [5] Katharopoulos, A., Vyas, A., Pappas, N., & Fleuret, F. (2020). Transformers are rnns: Fast autoregressive transformers with linear attention. In International Conference on Machine Learning.
>
> **Q2: I think many prior work (BigBird, ETC, Luna, Long short Transformers etc) have already shown that a combination of multiple sparse patterns or global components are useful. Along this line, I think the narrative of multiple attention being useful is not entirely new either.**
>
> **R:**  As mentioned in our clarification in response to Q1, Scatterbrain has a different goal compared to prior work: we study the attention approximation quality of sparse + low-rank, a fundamental matrix structure. Our main goal is not to simply argue that having multiple attention types is useful. As the reviewer points out, we build on a line of work with the theme of combining multiple types of attention. Our results in fact shed further light on why they work, as they are special cases of a single principled structure (which our work provides).
>
> Our focus on the classical and well-defined problem of matrix approximation, as opposed to simply designing an efficient model that performs well on downstream tasks (e.g., Longformer, Luna, Long-short transformer, etc.) affords us several advantages:
> - Easier understanding and theoretical analysis (Section 3, 4). We see that Scatterbrain yields an unbiased estimate of the attention matrix, and we can also understand how its variance changes.
> - Clear-cut evaluation based on approximation error, as well as the ability to directly replace a full attention layer with Scatterbrain attention without re-training (Section 5). This setting is increasingly important as transformer models are getting larger and training them from scratch has become prohibitively costly. Other methods such as Luna and Long-short transformer are not backward compatible with pre-trained models.
>
> Here we compare Scatterbrain with other work mentioned by the reviewer, showing how most of them are special cases of Scatterbrain. We will also add this discussion in the updated version of the manuscript.
> - Longformer: a special case of Scatterbrain where the sparse component is local attention, and the low-rank component is the global tokens. Global tokens can be considered a restricted form of low-rank approximation.
> - BigBird: a special case of Scatterbrain where the sparse component is local + random sparse attention, and the low-rank component is the global tokens. The use of global tokens makes the model unsuited for autoregressive modeling. On the other hand, Scatterbrain’s generality allows it to use other kinds of low-rank attention (e.g., Performer), and thus Scatterbrain works on both the causal/autoregressive and the bidirectional/non-causal attention settings. BigBird’s motivation is also quite different from ours: they aim to design efficient attention such that the whole Transformer model is still a universal approximator and is Turing complete. Our goal is more concrete and easier to evaluate: we approximate the attention matrices, to get a small Frobenius error between the Scatterbrain attention and the full attention matrices.
> - Luna (concurrent work, was uploaded to arXiv after the NeurIPS deadline): they use a fixed-length extra sequence and two consecutive attention steps: the context sequence attends to the extra sequence, and then the query sequence attends to the extra sequence. This is similar in spirit to low-rank attention (Linformer) and global tokens, but it is not a low-rank approximation due to the non-linearity between the two attention steps. It is not clear to us that it combines different kinds of attention.
> - Long-short transformer (concurrent work, was uploaded to arXiv after the NeurIPS deadline): a special case of Scatterbrain where the sparse component is local attention and the low-rank component is Linformer.

---

> > ### Author Response · Authors · 2021-08-10
> > **Additional Experiments**
> >
> > **Q3: The sparse variant here is Reformer, which is form of learnable sparsity. What about fixed sparse patterns, like Sparse Transformer or Longformer/Sparse Transformer?**
> >
> > **R:** As mentioned in the manuscript (line 235) and in Q1 (Generality), Scatterbrain is general enough to accommodate different kinds of sparse and low-rank approximations as its sub-components. In particular, we can combine Local attention / block sparse (from Sparse Transformer / BigBird) + Performer (instead of Reformer + Performer) in a similar fashion to Section 4.2. The support of the sparse matrix S will thus be fixed and not adaptive to input, but all the other steps are exactly the same.
> >
> > We have run additional experiments on the Local attention + Performer combination:
> >
> > ------------------
> > Experimental Details: Recall that in Section 5.1, we have shown Scatterbrain can reduce the attention memory of Vision Transformer by 98% at the cost of only 0.8% drop of accuracy when serving as a drop-in replacement for full attention without training on ImageNet. We show the results for local+performer variation with the same memory budget in the following table.
> >
> > | Attention | Top-1 Acc |
> > | ----------- | ----------- |
> > | Full Attention | 81.7%
> > | reformer (sparse) |  79.8%
> > | Local (sparse) |  79.6%
> > | Performer (low-rank) |  80.1%
> > | Scatterbrain (local+performer) |  80.3%
> > | Scatterbrain (reformer+performer)|  80.7%
> > ------------------
> >
> > Analysis: We see that:
> > - Both Scatterbrain variants outperform their own sparse and low-rank baselines, validating our theory that sparse+low-rank is strictly better than either just sparse or just low-rank
> > - The local combination also performs reasonably well (which achieves the same accuracy of BigBird shown in Q4 below), though not as well as the Reformer + Performer combination in the manuscript.
> >
> > We will further clarify this generality (that Scatterbrain can work for any pair of sparse and low-rank methods) in the updated manuscript.
> >
> > **Q4: For Long Range arena, Reformer and Performer are only reported. Given that there are other models evaluated in LRA, why did the authors not compare with the other models? For example, did the model outperform BigBird in the LRA paper?**
> >
> > **R:** The goal of our experiments is to validate our theory that Scatterbrain (sparse + low-rank) outperforms its sparse and low-rank baselines. As a result, in our experiments, we compared against the sparse baseline (Reformer) and the low-rank baseline (Performer), and did not compare against BigBird. More broadly, our main objective is not to get state-of-the-art performance on a specific benchmark, but to understand the strengths and weaknesses of two main attention approximation approaches (sparse and low-rank), and show how their combination yields strictly better results.
> >
> > However, after the reviewer raised the question, and we were also curious to see how BigBird performs (because it could be viewed as a special case in the Scatterbrain framework), we ran a quick additional experiment.
> >
> > -------------
> > Experimental Details: Recall that in Section 5.1, we have shown Scatterbrain can reduce the attention memory of Vision Transformer by 98% at the cost of only 0.8% drop of accuracy when serving as a drop-in replacement for full attention without training on ImageNet. We show the results for BigBird with the same memory budget in the following table.
> >
> > | Attention | Top-1 Acc |
> > | ----------- | ----------- |
> > | Full Attention | 81.7%
> > | reformer (sparse) |  79.8%
> > | Performer (low-rank) |  80.1%
> > | BigBird |  80.3%
> > | Scatterbrain|  80.7%
> > ------------------
> > Analysis: We can see that BigBird, as a special case of the general Scatterbrain framework(a single principled structure), also performs pretty well (only a little behind our suggested Scatterbrain variant). As we mentioned several times, our theoretical results shed further light on why they work. For completeness, we will add a more detailed comparison with other models on the LRA benchmark in the updated manuscript.
> >
> > **Q5: For language modeling, can the authors provide a strong local attention baseline? I think local attention should always at least be reported whenever xformer comes into play, given that it is the "simple naive baseline"**
> >
> > **R:** Thank you for this suggestion. We have added results (val perplexity) for local attention on the Copy and Wikitext-103 language modeling task:
> >
> > -------------
> > | Attention | Copy (ppl) | WikiText-103 (ppl) |
> > | ----------- | ----------- | ----------- |
> > | Full | 1 | 25.258
> > | Reformer | 6.8| 27.68
> > | Performer | 49| 66
> > | Local | 53| 30.72
> > | Scatterbrain | 2.58|26.72
> > -------------
> >
> > We see that local attention is reasonably competitive on Wikitext-103 but does not perform well on Copy. The results are not surprising as noted in the Reformer paper that Copy requires non-local attention lookups.

---

> > > ### Author Response · Authors · 2021-08-10
> > > **Other Detailed Questions**
> > >
> > > **Q6: The performance of Performer on Copy and Wikitext is horrible. Have the authors tuned the model carefully? I think the kernel choice in Performer is important, it would be good to know if Performer was adequately tuned.**
> > >
> > > **R:** We allocate the same tuning budget to all methods and use the FAVOR kernel that approximates softmax for fairness. Performer does not perform well on these datasets likely because the ideal attention pattern seems to be sparse rather than low-rank. For example, in the copy task, the input has the form 0w0w for some string w of length N, i.e. the model is supposed to “copy” the string w. Thus the ideal attention pattern is for each token to attend to exactly one other token N +1 steps before it. It is hard to get a low-rank approximation of this attention matrix, as it is full-rank with singular values of roughly the same magnitude. Given that Performer seeks a low-rank approximation of the full attention matrix, it is less suited for this task. Similarly, language modeling on Wikitext-103 requires attending to a few relevant tokens to predict the next token, making Performer (low-rank) less suited for this task.
> > >
> > >
> > > **Q7: Try combining different sparse and low-rank models, and draw insights from that. Are there benefits of different composition of attention types? What about different types of sparse models.**
> > >
> > > **R:** Choosing different sparse and low-rank sub-components allow Scatterbrain to easily trade off approximation accuracy and efficiency. For example, under the same memory budget, we observe Local attention + Performer is faster but slightly less accurate than Reformer + Performer.
> > >
> > > **Q8: What about interleaving layers? Does interleaving Reformer and Performer give on the necessary inductive bias as ScatterBrain?**
> > >
> > > **R:** We frame the attention approximation problem as matrix approximation, and thus focus on approximating the attention matrix of each layer independently. This framing allows us to easily analyze the tradeoff between accuracy and efficiency. It is not clear to us if interleaving sparse and low-rank layers will achieve the same goal, due to the much more complicated setting of approximating the outputs of multiple layers with nonlinearity in between. We hope to explore this in future work.

---

### Official Review · Reviewer_Gfk7 · 2021-07-17

**Rating:** 9
**Confidence:** 4

**Summary:**

This submission presents an efficient approximation method of attention matrices. This work demonstrates the effectivenss of the proposed method really well in the domain of dense attention, e.g., Transformer or self-attention.

The authors provided thorough motivational examples, both theoretical and empirical analyses, and discussions.

This work achieves a new trade-off point that works likely to be better than sparse or low-rank counterparts by using sparse + low-rank decomposition.
The proposed method maintains both favorable properties of Performer and Reformer, and extends the working regime of the algorithm by sacrificing a bit of cost over Reformer.

The developed algorithm is simple and exact in Supp(S). Also, the authors design the experiment very well, and show effectively that the proposed scatterbrain has lower approximation errors. In various tasks, it is shown that the proposed method maintains the efficiency of Reformer at a similar level, while extending the successful working regimes.

The proposed method is practical in that the pre-trained attention-based models can be sped up by replacing the attention computation as a drop-in replacement that maintains the original quality as much as possible.

I'm happy to be selected as a reviewer of this work. I'm enjoying reading this work, and appreciate the authors submitting high-quality work.

**Limitations And Societal Impact:**

The authors discussed the positive effect of the proposed method on efficiency and energy saving. Also, they warned about spreading fake data due to the advance of the generative models in multiple domains, e.g., NLP, image, and video, and its extension to privacy and fairness issues. They also argued the necessity of addressing those problems individually.

The discussion is reasonable and agreed upon.

**Main Review:**


Pros
- A new good trade-off between speed and accuracy is achieved
- Very well-motivated and thoroughly analyzed
- The proposed algorithm is well-designed such that it is easy to replace the pre-trained components of Transformer.

Cons
- I'm biased; thus, I couldn't find any major cons.
- Discussion about the rank hyper-parameter and its experiment is not discussed.


<Comments>

* This work mainly focuses on the efficiency improvement of the pre-trained attention-based models. However, in practice, the most bottleneck to adopt such large attention-based models is on training. This reviewer is aware of the fine-tuning experiment in the supplementary material, but it would be more impactful to demonstrate full training from scratch. This is likely to amplify the gap with Reformer or Performer (especially against Performer), because gradient efficiency of the scatterbrain is much higher.

* There must be a balance hyper-parameter between rank and sparsity. It acts as another trade-off controller. The discussion and phase transition on how to set the proper balance would be helpful to readers.



================= After rebuttal =================

The authors' responses answer this reviewer's questions and comments appropriately.
This reviewer keeps supporting the original rating.

**Time Spent Reviewing:**

5

---

> ### Author Response · Authors · 2021-08-10
> **Reply to Review by Reviewer Gfk7**
>
> We thank the reviewer for the strong support of our work! We appreciate your generous comments on our **thorough** theoretical & empirical analysis and **simple & exact** algorithm.
>
> **Q1: This reviewer is aware of the fine-tuning experiment in the supplementary material, but it would be more impactful to demonstrate full training from scratch.**
>
> **R:** We’ve presented the results of full-training-from-scratch experiments in Section 5.2. Specifically, we show Scatterbrain outperforms sparse or low-rank baselines on a wide range of tasks including, language modeling, classification, and Long-range Arena benchmarks.
>
> We agree with the reviewer that training-from-scratch-experiments are impactful and we’ve shown the practicality of our work through evaluation in multiple settings, such as inference, training, and fine-tuning. We are glad the reviewer also appreciated our fine-tuning experiments in the appendix!
>
> **Q2: There must be a balanced hyper-parameter between rank and sparsity. It acts as another trade-off controller. The discussion and phase transition on how to set the proper balance would be helpful to readers.**
>
> **R:** Yes, we agree that this hyper-parameter, the ratio between memory/parameter allocated to sparse and low-rank approximation, is an important implementation detail. We describe here some rules that we used in our experiments to set this ratio:
> - For inference, we set this ratio based on the entropy of an observed subset of attention matrices in different layers: we allocate more memory to the low-rank component compared to the sparse component if the entropy is high.
> - For training, generally allocating more memory budget to sparse tends to perform better, so in the experiment, we set the ratio to 3:1 (sparse: low-rank component) for simplicity.
>
> Moreover, in future work, it could be useful to make this ratio adaptive during training. For example, in the early stage of the training and early layers, attention matrices are usually more uniform (higher entropy). Thus, the approximation error could be even lower if the ratio favors low-rank-based components. One approach could be to monitor the approximation error of sparse and low-rank components compared to full attention regularly and adjust the memory budget accordingly. We will add the above discussion to the updated manuscript.

---

> > ### Comment · Reviewer_Gfk7 · 2021-08-19
> > **Response to the authors**
> >
> > Thanks for the rebuttal. This reviewer read the other reviewers' comments as well as the authors' responses.
> > I was satisfied with the authors' responses; thus, I keep my original rating.
> >
> > I have the following additional comments to improve the submission.
> > - I expected to see more detailed comparisons of full-training-from-the-scratch in a table form, not just a simple mention as done in Sec. 5.2. I could not find the experiment in the submission. If there were, please insert the reference link in Sec. 5.2.
> > - The provided information about the balance hyper-parameters would be very informative to the readers. Please include them in the final version. Also, this reviewer would like to suggest including a short ablation study in future work.

---

> > > ### Author Response · Authors · 2021-08-19
> > > **Full-training-from-the-scratch experiments location**
> > >
> > > Thanks again for your support!
> > >
> > > A quick clarification on where we put our full-training-from-scratch experiments:
> > > - They are on **Page 8-9** (Line 320-369) named **Section 5.2 End-to-end Training Performance** in our submission (main paper). The results are in **Table 2** and **Section 5.2.1 Auto-regressive Tasks** and **Section 5.2.2 Classification Tasks**.  We will add links to Section 5.2.1, 5.2.2 and Table 2 in Section 5.2 in our final version!
> > >
> > > Please let us know if you have further comments!

---

> > > > ### Author Response · Authors · 2021-08-19
> > > > **Full-training-from-the-scratch results**
> > > >
> > > > For your convenience, we copy our Table 2 below:
> > > >
> > > > -------------------------------
> > > > Table2: The performance of Scatterbrain, Reformer, Performer, and Full-Attention **training from scratch** on LongRange-Arena benchmarks and 2 popular language modeling tasks. We fix the same number of parameters (1/8 of the full) used for approximating the attention matrix for each method.
> > > >
> > > > -------------
> > > > | Attention | Copy (ppl) | WikiText-103 (ppl) |
> > > > | ----------- | ----------- | ----------- |
> > > > | Full | 1 | 25.258
> > > > | Reformer | 6.8| 27.68
> > > > | Performer | 49| 66
> > > > | Scatterbrain | **2.58**| **26.72**
> > > > -------------
> > > >
> > > > | Attention | ListOps | Text | Retrieval | Image | Pathfinder | Avg Accuracy|
> > > > | ----------- | ----------- | ----------- | ----------- | ----------- | ----------- | ----------- |
> > > > | Full Attention | 38.2 | 63.29 | 80.85 | 41.78 | 73.98 | 59.62
> > > > | Reformer | 36.85 | 58.12 | 78.36 | 28.3 | 67.95 | 53.9
> > > > | Performer | 35.75 | 62.36 | 78.83 | 39.71 | 68.6 | 57.05
> > > > | Scatterbrain | **38.6** | **64.55** | **80.22** | **43.65** | **69.91** | **59.38**

---

### Decision · Program_Chairs · 2021-09-27

**Decision:**

Accept (Poster)

**Comment:**

In this paper, the authors discussed the effect of sparse and low-rank approximation for the transformer. The proposed combination leads to better performances comparing to the existing fast transformer with each approximation scheme only.  The paper is well-motivated and the theoretical and empirical analyses are reasonable. Therefore, I recommend the accept this submission.

The paper can be further improved by taking the reviewers' suggestions about the experiments, e.g., full-training-from-the-scratch  (Reviewer Gfk7), comparison on LRA (6aFS), tradeoff between memory vs. accuracy (QK3S), and comparing with more baselines (Hndw).